# Symmetric Kernels with Non-Symmetric Data: A Data-Agnostic Learnability Bound

## Abstract

Kernel ridge regression (KRR) and Gaussian processes (GPs) are fundamental tools in statistics and machine learning, with recent applications to highly over-parameterized deep neural networks. The ability of these tools to learn a target function is directly related to the eigenvalues of their kernel sampled on the input data. Targets that having support on higher eigenvalues are more learnable. While kernels are often highly symmetric objects, the data is often not. Thus, kernel symmetry seems to have little to no bearing on the above eigenvalues or learnability, making spectral analysis on real-world data challenging. Here, we show that contrary to this common lure, one may use eigenvalues and eigenfunctions associated with highly idealized data measures to bound learnability on realistic data. As a demonstration, we give a theoretical lower bound on the sample complexity of copying heads for kernels associated with generic transformers acting on natural language.

## 1 Introduction

Gaussian process regression (GPR) and kernel ridge regression (KRR) are workhorses of statistics and machine learning. GPR and KRR are also intimately related - given the same kernel function, they both result in the same predictor (Rasmussen & Williams, 2006; Kimeldorf & Wahba, 1970). More recently, several correspondences have appeared between GPs and kernel methods with deep neural networks (DNNs) have appeared (Lee et al., 2018; Jacot et al., 2018; Matthews et al., 2018; Novak et al., 2018; Cohen et al., 2021). Perhaps the most striking result (Novak et al., 2018; Lee et al., 2020) is that GPs drawn from the kernels of CNNs exhibit quite an impressive performance, which does not fall much behind that of those CNNs. In addition, the infinite width/channel/attention-heads limit has served as a starting point for several extensions to finite DNNs in the feature-learning regime (Li & Sompolinsky, 2021; Seroussi et al., 2023; Ariosto et al., 2022). Thus, apart from the direct interest in GPs and kernel methods, predicting what GPs and kernels can learn appears as a stepping stone to predicting neural networks' learnability.

The complexity of computing with GPs or kernels and the analytical difficulty of reasoning about them both stem from the need to invert a large matrix associated with the kernel between each pair of data points (Rasmussen & Williams, 2006). Various theoretical studies indicate that the main objects controlling learnability are the eigenvalues and eigenfunctions associated with this matrix in the infinite data limit (Silverman, 1984; Sollich & Williams, 2004; Canatar et al., 2021b; Bordelon et al., 2021; Simon et al., 2023). In this eigen-learning framework, the regression target is decomposed to eigenfunctions and the regression roughly filters out eigenfunctions with eigenvalues below $\sigma^2/P$, $P$ being the amount of data and $\sigma^2$ the ridge parameter or an effective ridge parameter. Notwithstanding, this approach still leaves us with the formidable task of diagonalizing the kernel on the data measure. In addition to being a hard computational task, it also requires an accurate understanding of the underlying data measure which, apart from specialized use cases such as physical-informed neural networks (Raissi et al., 2019), is out of our grasp. Thus, while kernels show much promise as indicators of DNN performance, leveraging them to provide insights on how DNNs perform on real-world data remains a challenge.

Drawing inspiration from physics, symmetries can greatly simplify diagonalization processes, using tools from representation theory. Conveniently, kernels associated with DNNs are often quite symmetric. For instance, any kernel associated with a fully-connected-network (FCN) (resp. transformer)

would have rotation (e.g. (Basri et al., 2019; Yang & Salman, 2020; Scetbon & Harchaoui, 2021; Bietti & Bach, 2021)) (resp. permutation (Lavie et al., 2024)) symmetry in input space. Thus, if acting on similarly symmetric data, representation theory arguments can be directly applied to diagonalize/block-diagonalize kernels (Fulton & Harris, 2004; Tung, 1985). As an example, for data uniform on the hypersphere, any FCN kernel is diagonalized by hyper-spherical harmonics. Furthermore, the finite sum rule on eigenvalues in conjugation with degeneracies implied by symmetry further forces strict upper bounds on the eigenvalues (Cohen et al., 2021). Unfortunately, real-world data is rarely uniform or symmetric. Consequently, the eigenvalue problem, involving both the kernel and the data measure, loses its symmetry properties and kernel-symmetry alone does not help one obtain the kernel spectrum required for estimating learnability.

In this work, we provide a readily computable lower bound on the number of samples needed to learn idealized target features on real-world datasets using Gaussian processes. Central to our bound is the use of an auxiliary test distribution $(q(x))$, invariant under symmetries of the kernel, to measure learnability. We show that such $q(x)$-learnability (or cross-dataset learnability) can be bounded from above without ever solving the difficult kernel eigenvalue problem on the real data. Instead, our bound depends on two relatively accessible quantities: (**1**) The eigenfunctions and eigenvalues of the kernel on $q(x)$ and (**2**) The norm of the eigenfunctions associated with the target on the real dataset (typically an $O(1)$ number). As we demonstrate, since the eigenvalue problem w.r.t. $q(x)$ enjoys all kernel symmetries, it is largely tractable using representation theory tools. We further find that our $q(x)$-based measure of learnability correlates well with that on the training set.

**Our main contributions are**:

- We generalize the definition of learnability to a scenario where the training is done on one dataset, but the learnability is tested on another (cross-dataset learnability). We provide numerical evidence that this, more tractable, learnability correlates well with the standard one.
- We prove a sample complexity lower bound for kernel ridge regression and Gaussian process regression on real-world datasets based on our cross-dataset learnability and show it gives prediction experimentally.
- We study several examples of relevant architecture, including transformers, discuss their symmetries, and use representation theory and the above bound to derive concrete sample complexity lower bounds.
- Our bound is a prescription to carry over sample complexity results found in the literature that builds upon idealized datasets (Basri et al., 2019; Lavie et al., 2024; Bietti & Bach, 2021; Azevedo & Menegatto, 2014; Scetbon & Harchaoui, 2021; Yang & Salman, 2020) to real-world datasets in the form of lower bounds.

The paper is organized as follows. The setting and assumptions are presented in Section 2, followed by a statement of the main result in Section 3 together with experimental results that show its usefulness on real-world datasets. We then move on to a series of vignettes, each one presents a scenario where the bound is used and discusses certain aspects of the result. The first vignette 4.1 uses a linear regression setting with Gaussian i.i.d. data to showcase a simple implementation of the result and discuss how the details of the specific training distribution are accounted for in the bound. The second vignette 4.2 examines a fully connected network that learns parity on the hyper-cube with a correlated measure and discusses measuring learnability out-of-distribution. The last vignette 4.3 presents a practical result, bounding from below the sample complexity of copying heads for transformer architecture. Finally, we provide an outlook in Section 5.

## 2 SETTING

In this section, we present a short introduction to kernel ridge regression (KRR) or Gaussian process regression (GPR) and the concept of learnability, followed by a generalization to cross-dataset scenarios.

The regression setting includes a kernel function $k(x, y)$, a ridge parameter[1] $\sigma^2$, and a dataset $D_P = \{(x_\mu, y(x_\mu))\}_{\mu=1}^P$ of $P$ data points with $x_\mu$ being the $\mu$'th input and $y(x_\mu)$ being the regression

---

[1]or an effective ridge parameter as a (as is the case for neural tangent kernel) (Canatar et al., 2021b)

target/label for $x_\mu$. In this case, the predictor is given by[2]

$$\hat{f}_D(x) = \sum_{\nu,\rho=1}^{P} k(x, x_\nu) \left[K + I\sigma^2\right]_{\nu\rho}^{-1} y(x_\rho); \quad [K]_{\mu\nu} = k(x_\mu, x_\nu), \tag{1}$$

where $I$ is the identity matrix and $x_\mu \in D_P$. This expression can be rewritten in the eigenbasis of the kernel

$$\hat{K}_D \psi_i(x) := \mathbb{E}_{x' \sim D_P} \left[k(x, x')\psi_i(x')\right] = \lambda_i \psi_i(x), \;\; \mathbb{E}_{x \sim D_P} \left[\psi_i(x)\psi_j(x)\right] = \delta_{ij}, \;\; x \in D_P, \tag{2}$$

with $\delta_{ij}$ the Kronecker delta and $\mathbb{E}_{x \sim D_P}[\cdot]$ being expectation value w.r.t the dataset $D_P$, yielding

$$\hat{f}_D(x) = \sum_i \frac{\lambda_i}{\lambda_i + \sigma^2/P} \psi_i(x) \frac{1}{P} \sum_{\mu=1}^{P} \psi_i(x_\mu) y(x_\mu). \tag{3}$$

The learnability of the feature $\psi_t$ (eigenvector of the kernel) (Simon et al., 2023)

$$[0,1] \ni \mathcal{L}_t := \left| \frac{\mathbb{E}_{x \sim D_P}\left[\psi_t(x)\hat{f}_D(x)\right]}{\mathbb{E}_{x \sim D_P}\left[\psi_t(x)y(x)\right]} \right| = \frac{\lambda_t}{\lambda_t + \sigma^2/P} \tag{4}$$

can be identified in Eq.(3), as a measure for the degree to which the regression can reconstruct the eigenvector coefficient from the target function. This is a simplified version of learnability which was introduced in (Simon et al., 2023; Canatar et al., 2021b) which builds on the eigenvalue decomposition on the underlying dataset measure rather than the empirical one and allows to extend the results to the test set by replacing $\sigma^2$ with an effective one.

When the learnability is close to unity, the coefficient is copied perfectly and we say the feature is learned, while a vanishing learnability means the predictor cannot use the feature at all and we say it is not learned. Learnability results can be restated as a function of the number of samples, resulting in sample complexity; e.g. requiring learnability is $1 - \epsilon$

$$\mathcal{L}_t \overset{!}{=} 1 - \epsilon \Rightarrow P_t^* = \lambda_t^{-1} \sigma^2 \frac{1 - \epsilon}{\epsilon}, \tag{5}$$

with $P^*$ the sample complexity to achieve learnability that is $\epsilon$ close to unity.

We consider here a cross-dataset generalization of this setting, where one solves the eigenvalue problem given in Eq.(2) on an auxiliary dataset with a probability measure $q(x)$, but performs the regression on the dataset $D_P$.

$$\mathcal{L}_t^{D,q} := \left| \frac{\mathbb{E}_{x \sim q}\left[\phi_t(x)\hat{f}_D(x)\right]}{\mathbb{E}_{x \sim q}\left[\phi_t(x)y(x)\right]} \right| \tag{6}$$

with $\phi_t$ eigenfunction of the kernel w.r.t $q(x)$, namely it solves Eq. (2) with $D_P$ replaced by $q$. This quantity is simply the ratio between the magnitude of the component $\phi_t$ in the predictor to what it should be to reconstruct the target function perfectly.

Comparing equations (4),(6) the inner product (equivalently expectation) in the cross-data learnability is taken over the auxiliary measure instead of on the dataset. This change can be intuitively understood as learning from $D_P$ but judging how good the reconstruction is based on functional similarity on $q(x)$. Such a change can have the advantage of weighting all possible inputs evenly, perhaps capturing a notion of out-of-distribution generalization but the disadvantage of being uninformed about the details of the specific dataset. The issue of judging similarity differently based on different measures is exemplified and discussed further in Section 4.2.

Cross-dataset learnability agrees with the common one (Simon et al., 2023) when learning from the true underlying probability density of the dataset $p(x)$ (rather than the empirical $D_P$) and choosing $q(x) = p(x)$. However, The underlying density $p(x)$ is often inaccessible[3], in these cases, one can still use our generalized learnability to estimate learnability from the empirical dataset $D_P$.

---

[2]Note that the resulting predictor from KRR with a kernel function $k(x, y)$ and ridge $\delta$ is identical to GPR with covariance function $k(x, y)$ and observation uncertainty $\sigma^2 = \delta$ (Kimeldorf & Wahba, 1970).

[3]That is, one does not have access to the "platonic" description of the data.

We note that while the on-dataset learnability in Eq. (4) is bounded $\mathcal{L}_t \in [0,1)$, the cross-dataset learnability is unbounded from above. As a consequence, maximizing cross-dataset learnability does not imply good learning; instead, one must require it to be close to unity.

In a contemporary context, a kernel of particular interest is the neural tangent kernel (NTK) which describes an NN trained with gradient flow (Jacot et al., 2018). A second example is the neural network Gaussian process (NNGP) which describes Bayesian inference with a prior induced by the distribution of the NN weights at initialization (Neal, 1996; Lee et al., 2018) or when training a NN with noisy gradients (Naveh et al., 2021; Welling & Teh, 2011).

## 3 RESULT

**Theorem 3.1.** *Given a GP kernel $k(x,y)$ and its eigendecomposition $\{(\lambda_i, \phi_i(x))\}$ on an auxiliary probability density (measure) $q(x)$ and a dataset $D_P = \{(x_\mu, y(x_\mu))\}_{\mu=1}^P$ of $P$ samples, the cross-dataset learnability (see Eq.6) is bounded from above by*

$$\mathcal{L}_t^{D,q} \leq \sigma^{-2} \lambda_t P \frac{\sqrt{\mathbb{E}_{x \sim D_P}\left[\phi_t^2(x)\right] \mathbb{E}_{x \sim D_P}\left[y^2(x)\right]}}{\left|\mathbb{E}_{x \sim q(x)}[\phi_t(x)y(x)]\right|}. \tag{7}$$

*Therefore, the number of samples $P^*$ required to achieve cross-dataset learnability $\mathcal{L}_t^{D,q} = 1 - \epsilon$ for a specific target feature $\phi_t(x)$ is bounded from below by*

$$P^* \geq \sigma^2 \lambda_t^{-1}(1-\epsilon) \frac{\left|\mathbb{E}_{x \sim q(x)}[\phi_t(x)y(x)]\right|}{\sqrt{\mathbb{E}_{x \sim D_P}\left[\phi_t^2(x)\right] \mathbb{E}_{x \sim D_P}\left[y^2(x)\right]}}, \tag{8}$$

*with $\mathbb{E}_{x \sim D_P}[\cdot]^4$ ($\mathbb{E}_{x \sim q(x)}[\cdot]$) being expectation value w.r.t the dataset $D_P$ ($q(x)$). The bound holds as long as the training dataset $D_P$ is a subset of the support of $q(x)$*

$$\text{supp}(q) \supseteq \{x\}_{\mu=1}^P; \tag{9}$$

*note that this is the only requirement on $q(x)$ and one may choose the most favorable one within this class. The proof is given in Appendix A.*

The result in Eq.(8) can be interpreted as follows. At least $P^*$ samples are required to learn a feature $\phi_t(x)$ from the dataset $D_P$ based on the corresponding eigenvalue found by performing eigenvalue decomposition of the kernel on $q(x)$. It can be deployed whenever one is able to perform the eigenvalue decomposition on $q(x)$ more easily than on $D_P$, for $q(x)$ that satisfies Eq.(9).

This bound can be used as a prescription for carrying over sample complexity results found on a favorable auxiliary measure $q(x)$ to a rich dataset of interest $D_P$. One has to perform an eigenvalue decomposition for the kernel operator $\hat{K}_q$ on $q(x)$ and plug the result into Eq.(8). Three additional quantities are needed the overlap between the target $y(x)$ and the feature of interest on $q(x)$ $\mathbb{E}_{x \sim q(x)}[\phi_t(x)y(x)]$; the norm of the feature of interest $\phi_t$ on $D_P$ $\mathbb{E}_{x \sim D_P}\left[\phi_t^2(x)\right]$; and the norm of the target $y$ on $D_P$ $\mathbb{E}_{x \sim D_P}\left[y^2(x)\right]$. *Comparison with other measures of learnability.* Our bound (8) can be compared with a common learnability measure (Simon et al., 2023)[5] that can be deployed given a solution of the generically intractable eigenvalue problem on the non-symmetric data distribution

$$\mathcal{L}_{t_*} = \frac{\eta_{t_*}}{\eta_{t_*} + \sigma^2/P}, \tag{10}$$

With $\eta_{t_*}$ being the eigenvalue for the feature of interest found by solving the eigenvalue problem on the training distribution that underlies $D_P$. Requiring learnability $\mathcal{L}_{t_*} = 1 - \epsilon$ we find

$$1 - \epsilon = \frac{\eta_{t_*}}{\eta_{t_*} + \sigma^2/P} = \frac{\eta_{t_*} P}{\sigma^2} + O\left(\left(\frac{\eta_{t_*} P}{\sigma^2}\right)^2\right) \leq \frac{\eta_{t_*} P}{\sigma^2}, \tag{11}$$

---

[4]This quantity may naturally fluctuation a little as one scans the values of $P$.

[5]Note that our notation slightly differs from (Simon et al., 2023) as we take out the scaling of the effective ridge with P, as done in (Canatar et al., 2021b; Cohen et al., 2021).

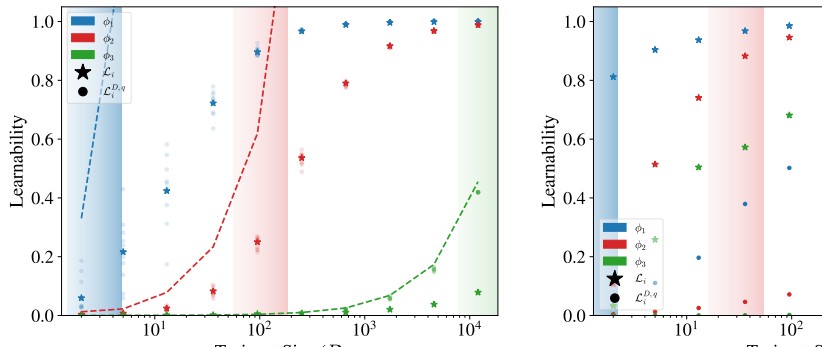

Figure 1: **Left: (The onset of learnability is tightly bounded in an idealized setting)** The learnability computed empirically (stars) the cross-dataset learnability (dots) and our bound on the cross-dataset learnability (dashed) of a random linear $\phi_1$, quadratic $\phi_2$ and cubic $\phi_3$ target features. The trainset consists of $10^4$ samples drawn uniformly on the hypersphere $\mathbb{S}^{13}$ and $q(x)$ is a uniform (continuous) distribution on the hypersphere. The shaded areas indicate a learning region, given by our bound taken at equality for $\epsilon = 0.7$ and until $\epsilon = 0$. The bound is seen to be tight before and around the onset of learning even for a single realization. Notably, we do not expect the bound to be tight when the feature is already learned well, but to predict the minimum required number of samples for learning. **Right: (Cross-dataset learnability predicts sample complexity on Real Datasets)** same as the figure to the left but with *Fashion MNIST* as the dataset $D_P$. The shaded learning regions give a good indication of when the features will be learned. Cross-dataset learnability is seen to be correlated with the dataset learnability in an imperfect manner, this is the price paid for the huge simplification in estimating the learnability. See also Fig. 2 below.

where the last inequality arises because the learnability is a concave function of $P$. We arrive at a similar inequality as a condition for learnability but with a different pair of eigenvalue-eigenfunction $\eta_t, \varphi_t$. $\varphi_t$ is guaranteed to be normalized as the eigendecomposition was carried out with the same probability measure as the dataset. From this perspective, the bound can be expected to be tight when $P \ll \eta_{t_*}^{-1}\sigma^2$ and less tight as $P$ grows larger. This is coherent with the view of our bound as a *necessary* condition for learnability, rather than an exact prediction of the learnability. The power expansion approximates the learnability well when far from good learnability and bounds it from above throughout training.

In Figure 1, left panel, we show an experiment of exact KRR with a kernel that corresponds to a single hidden-layer ReLU network learning random linear, quadratic, and cubic target features $y = \phi_t$ for randomly chosen hyperspherical harmonics $\phi_1, \phi_2, \phi_3$ respectively. The dataset $D_p$ is $10^4$ samples drawn uniform i.i.d. on the hypersphere $\mathbb{S}^{13}$ ($d = 14$). The symmetric auxiliary measure $q(x)$ is naturally chosen to be the underlying symmetric distribution, (continuous) uniform on the hypersphere. We plot both the on-dataset learnability as in Eq. (4) (stars) and the cross-dataset learnability as in Eq. (6) (dots) which here is just the full underlying distribution. In this case, the bound on the learnability is seen to approximate the beginning of the learning stage well. The regime in which the bound is tight indicates an important feature of our result. It captures the onset of learning well and thus can be used to judge sample complexity, but misses the saturation of the learnability at later stages of learning. This view is further enforced by the visualization of the sample complexity bound which marks with shades the region where our sample complexity bound holds as an equality for $\epsilon \in [0.7, 0]$. We stress that the dots indicate a single random realization of the dataset, and the bound is guaranteed to hold for every such realization.

Figure 1, right panel, shows the same single hidden layer ReLU network KRR learning the same features from two classes of Fashion MNIST dataset (Xiao et al., 2017). The input dimension is reduced by PCA to $d = 14$, capturing $\approx 90\%$ of the variance in the dataset, as a balance between the dimension and the number of samples that were empirically found to allow learning of cubic features[6]. Afterward, each sample is normalized such that $\forall x \in D_p \|x\|_2^2 = 1$. We again compare the on-dataset learnability (stars) to the cross-dataset learnability which uses the hypersphere as

---

[6]We note that this choice also approximately matches the intrinsic dimension measured for Fashion MNIST (Aumüller & Ceccarello, 2021)

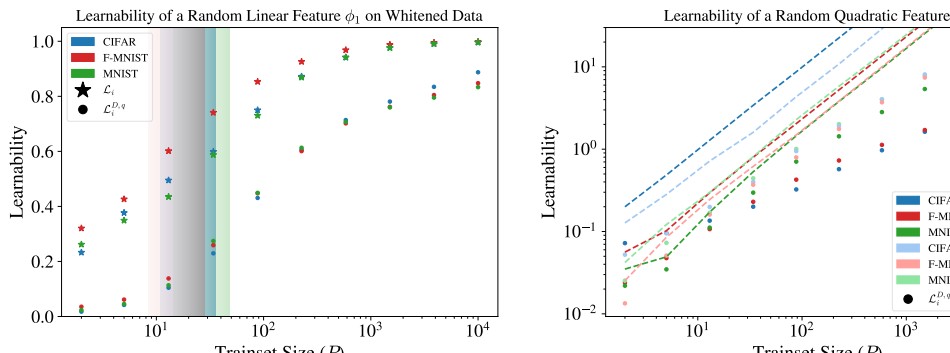

Figure 2: **Left: (Increasing dataset ($D_P$) and auxiliary measure ($q(x)$) similarity improve similarity between learnability and cross-dataset learnability)** Learnability (stars) and cross-dataset learnability (dots) of a linear hyperspherical harmonics are shown alongside an indicator for the sample complexity bound. Different datasets are shown in different colors, the datasets are PCA-whitened. The cross-dataset learnability is well correlated with the on-dataset learnability, showing that when $D_P, q(x)$ are closer together the learnabilities are also closer together. **Right: (Increasing dataset and auxiliary measure similarity yield a tighter bound)** This figure uses a setting similar to the one on the left but with a quadratic hyperspherical harmonic. Results for the whitened datasets are presented in lighter (whitened) color, while regular PCA is in full color. The bound on the cross-dataset learnability is plotted in a dashed line. Again using more similar distributions is beneficial and the bound is seen to be tighter in this case.

the auxiliary measure. The shaded learning regions determined by our sample complexity bound again give a good indication of how much data is required to learn a feature based on the regular learnability. As expected, our bound is more precise on the cross-dataset learnability and gives accurate predictions for where the learning starts.

Given the discussion on the relation between cross-dataset learnability and regular learnability one may wonder whether choosing an auxiliary measure $q(x)$ which is closer to the dataset $D_P$ would give better results; the experiments in Fig. 2 give an affirmative answer. In these experiments we process the three datasets, CIFAR-10 (Krizhevsky, 2009), Fashion MNIST (Xiao et al., 2017) and MNIST (LeCun et al., 2010), as above but PCA *whitening* and $d = 20$[7]. PCA-whitening makes the data covariance spherically symmetric, and hence makes it more similar to the hypersphere. The results in the left panel of Fig. 2 indeed show the cross-data set learnability (dots) follows the trends in the regular learnability (stars) closely. The whitening process also improves the tightness of our cross-dataset learnability bound as seen in the right panel. In light (whitened) color indicates whitened data and full color indicates non-whitened data, the bound (dashed line) for the whitened data is consistently tighter showing that a closer $D_P, q(x)$ pair results in a tighter bound.

### 3.1 Symmetries and the auxiliary measure $q(x)$

Here, we motivate choosing the auxiliary distribution $q(x)$ based on guiding principles of symmetry. We then present a workflow for using the result in Eq. (8) in conjunction with representation theory to bound the sample complexity of KRR and GPR on real-world data.

**Definition 3.2** (Kernel Symmetry). We say a kernel $k(x, y)$ has a symmetry group $G$ if

$$\forall g \in G \quad k(T_g x, T_g y) = k(x, y) \tag{12}$$

for $T_g$ a faithful representation of $g \in G$ *Intuitively*, a kernel symmetry means the kernel treats a pair of inputs and their symmetry augmented pairs in the same way.

**Definition 3.3** (Dataset Symmetry). We say a dataset measure $q(x)$ has a symmetry group $G$ if

$$\forall g \in G \quad q(T_g x) = q(x) \tag{13}$$

for $T_g$ a faithful representation of $g \in G$. *Intuitively*, a dataset symmetry means a symmetry-augmented version is as likely to be seen in the dataset as the original one.

---

[7]Again approximately matching the intrinsic dimension of CIFAR-10 Pope et al. (2021)

When both the kernel and dataset respect a symmetry group $G$ we say it is a symmetry of the kernel *operator*. A symmetry of the kernel operator can be used to asymptotically bound the eigenvalues from above using the dimension of their corresponding irreducible representations (irreps), as shown for the case of fully connected networks (Basri et al., 2019; Yang & Salman, 2020; Scetbon & Harchaoui, 2021; Bietti & Bach, 2021) and transformers (Lavie et al., 2024). A natural choice for $q(x)$ is therefore one that respects all the kernel symmetries.

$$\forall g \in G \quad k(T_g x, T_g y) = k(x, y) \rightarrow q(T_g x) = q(x). \tag{14}$$

Using the result in Eq.(8) and symmetries a learnability bound can be found using the following workflow:

1. Identify the kernel symmetries.

2. Define a relevant distribution $q(x)$ that has all the symmetries of the kernel and includes the training dataset in its support.

3. Decompose the target function into irreducible representations of the symmetry group.

4. Use the inverse of the dimension of the irrep[8] as an upper bound for the corresponding eigenvalue.

5. Use the eigenvalue bound together with Eq.(8) to find a sample complexity lower bound.

In Section 4 we present examples that use this workflow.

## 4 VIGNETTES

### 4.1 AWARENESS TO DATASET & LINEAR REGRESSION ON A LOW DIMENSIONAL DATA MANIFOLD

The purpose of this example is twofold, first, present a simple setting as an example of how one can use our main result, and second, discuss the ways in which our bound accounts for different training datasets.

Consider the kernel $K(x, y) = \frac{1}{d} x \cdot y$ with $x, y \in R^d$ and $q(x) = \prod_{i=1}^{d} p_N(x_i)$ where $p_N(x_i)$ are standard centered Gaussians. Let $D_P$ consist of $P$ $d$-dimensional vectors sampled i.i.d. such that for each $\vec{x} \in D_P$

$$\vec{x} = (x_1, x_2, ..., x_d), \quad x_1 \sim N(0, d), x_{i>1} = 0. \tag{15}$$

Where the scale was chosen such that the sum of eigenvalues (trace) of the kernel on both measures equals unity

$$\int k(\vec{x}, \vec{x}) \frac{e^{\frac{x_1^2}{2d}}}{\sqrt{2\pi d}} \prod_{i=2}^{d} \delta(x_i) \, d\vec{x} = \int k(\vec{x}, \vec{x}) \prod_{i=1}^{d} \frac{e^{\frac{x_i^2}{2}}}{\sqrt{2\pi}} \, d\vec{x} = 1 \tag{16}$$

with $\delta(x)$ the Dirac delta distribution, such that the learnability budget is the same in both distributions. Finally, let $y(\vec{x}) = x_1$.

Let us first estimate the learnability here without using the above bound. The kernel on this dataset coincides with $d^{-1} x_1 \cdot x_1$. From an equivalent kernel perspective, this has a single non-zero eigenvalue $\lambda$, associated with the function $x_1/\sqrt{d}$, given by

$$\int_{-\infty}^{\infty} dx_1 \frac{y_1 x_1}{d} \frac{x_1}{\sqrt{d}} \frac{e^{\frac{x_1^2}{2d}}}{\sqrt{2\pi d}} = \lambda \frac{y_1}{\sqrt{d}}; \qquad \lambda = 1 \tag{17}$$

hence standard learnability implies that $P^* = \lambda^{-1} \sigma^2 = \sigma^2$, for $\epsilon = 1/2$, the crossover value allowing us to learn half the target value.

---

[8]when the kernel is normalized such that $\mathbb{E}_{x\ q}[k(x, x)] = 1$, otherwise one has to multiply the result by $\mathbb{E}_{x\ q}[k(x, x)]$

Next, we apply our bound. To this end, we require the eigenfunctions and eigenvalues with respect to $q(x)$. These can be checked to be

$$\phi_i(x) = x_i \qquad \lambda_i = d^{-1} \tag{18}$$

Our target in these terms is $y(x) = \phi_1(x)$. Applying the bound in equation (8) for $\epsilon = 1/2$

$$P^* = \frac{\sigma^2 \lambda_1^{-1}}{2} \frac{\mathbb{E}_{x \sim q(x)}[\phi_1(x)y(x)]}{\sqrt{\mathbb{E}_{x \sim D_P}[\phi_1^2(x)]\,\mathbb{E}_{z \sim D_P}[y^2(x)]}} \approx \frac{\sigma^2}{2}\frac{d}{d} = \frac{\sigma^2}{2} \tag{19}$$

where the last approximation is due to replacing empirical sampling ($D_P$) with the expected one. In this case, the resulting bound is seen to be within a factor of 2 from the exact result, but in general, the tightness of the bound remains to be studied further.

*Awarness to the training dataset.* Clearly, learning the target from the symmetric measure $q(x)$ is a harder task, requiring $P^* = \sigma^2 d$ samples. We see our bound (8) encodes the information about the training dataset $D_P$ solely by the expected norms $\sqrt{\mathbb{E}_{x \sim D_P}[\phi_1^2(x)]}$, $\sqrt{\mathbb{E}_{x \sim D_P}[y^2(x)]}$ of the target function and feature; which scales with $d$ in the example above.

## 4.2 Measures of Learnability & Learning Parity with a Correlated Dataset

We continue by recovering a known result - learning parity from a uniform distribution on the hyper-cube with an FCN-GP is hard (Yang & Salman, 2020; Simon et al., 2023) and extend it to general distributions on the hyper-cube. We then present an example where the data measure has very low entropy and probes only a small low dimensional space of the hyper-cube. In that example, a function that mostly agrees with parity on the data can be learned easily, but learning a predictor that generalizes out-of-distribution remains hard. We discuss what learning a function can mean when considering different distributions, potentially including out-of-distribution test points, and suggest maximal entropy distributions as a reasonable measure upon which learnability can be gauged. Finally, we compare the suggested measure of learnability to a familiar one.

Consider learning parity on the $\vec{x} \in \{-1,1\}^d$ hyper-cube using FCNs. Here $\vec{x}$ is drawn from an arbitrary, possibly correlated, measure on the hyper-cube, and the target function is parity $y = \Pi_{i=1}^{d} x_i$ with no added noise.

Learning parity with noise is believed to require $P$ scaling exponentially with $d$ (Blum et al., 2000). Parity without noise can be learned with $O(d)$ samples using Gaussian elimination and relations between boolean operations and algebra in $Z_2$ fields (Blum et al., 2000). A GP, however, involves a larger hypothesis class including real rather than boolean variables. It also seems highly unlikely that it could learn from examples an $O(d^3)$-algorithm such as Gaussian elimination. In the case of a uniform measure, an FCN GP is known to require $P^*$ which is exponential in $d$ (Simon et al., 2023) to learn parity. It is reasonable to assume that a generic non-uniform measure would not reduce the complexity of this task, however, we are not aware of any existing bounds applicable to this broader case.

To this end, we take as an ideal distribution $q(x)$ a uniform distribution on the sphere containing the corners of the hyper-cube. We turn to calculate or bound the different elements in Eq. 8. First, we require the $\phi_t(x)$ associated with parity. Under $q(x)$, any FCN kernel is diagonal in the basis of hyperspherical harmonics. The latter can be described as rank-$n$ homogeneous harmonic polynomials (Frye & Efthimiou, 2012). Each rank constitutes an irreducible representation of the rotation group (Tung, 1985; Fulton & Harris, 2004). As a consequence, each rank-$n$ polynomial is a kernel eigenfunction whose eigenvalue depends only on $n$. We may conclude our eigenfuction of interest $\phi_t$ is $\hat{n}^{-1/2} y$ where $\hat{n}$ is a normalization factor w.r.t. to the measure $q(x)$. Having identified the eigenfunction we would like to estimate its eigenvalue. One can show that there are $N(n,d) = \frac{2n+d-2}{n}\binom{n+d-3}{n-1}$ polynomials at given $n, d$ (Frye & Efthimiou, 2012). Considering parity, it is a harmonic homogeneous polynomial of rank $d$ and consequently part of a $N(d,d)$-degenerate subspace of any FCN kernel with eigenvalue $\lambda_d$. Noting that $\mathbb{E}_q[K] = \int dx\, q(x)K(x,x)$ equal the sum of all eigenvalues and that eigenvalues are positive, we obtain $\lambda_d \leq \mathbb{E}_q[K]/N(d,d)$.

Following its appearance in the numerator and denominator and the fact that parity squares to 1 on the hyper-cube we find

$$P^* \geq \sigma^2 \lambda_t^{-1}(1-\epsilon)\frac{\mathbb{E}_{x \sim q(x)}[y(x)y(x)]}{\sqrt{\mathbb{E}_{x \sim D_P}[y^2(x)]\,\mathbb{E}_{z \sim D_P}[y^2(x)]}} \geq \sigma^2(1-\epsilon)\hat{n}N(d,d)/\mathbb{E}_q[K] \tag{20}$$

We calculate the normalization factor $\hat{n}$ in Appendix D, and quote the result here

$$\hat{n} = \frac{2^{-d} d^d \Gamma\left(\frac{d}{2}\right)}{\Gamma\left(\frac{3d}{2}\right)}. \tag{21}$$

Next, we use Stirling's formula for an asymptotic expansion for $d \gg 1$

$$P^* \geq \frac{\sigma^2(1-\epsilon)}{\mathbb{E}_q[K]} \sqrt{\frac{3^3}{2^6 \pi d}} \left(\frac{2^4 e^2}{3^3}\right)^{\frac{d}{2}} \approx \frac{\sigma^2(1-\epsilon)}{\mathbb{E}_q[K]} \sqrt{\frac{3^3}{2^6 \pi d}} (2.09...)^d. \tag{22}$$

We find that in high-dimension $d \gg 1$, given any training dataset on the hyper-cube, the sample complexity of parity for a FCN GP is at least exponential in $d$.

An extreme yet illustrative case to consider is a predominantly correlated measure on the hyper-cube which forces all $x_i$'s to be equal $p_1(x)$ plus a uniform measure namely $p(x) = (1-\alpha)p_1(x) + \alpha q(x)$ with $\alpha \ll 1$. For even $d$ on such measure, $y$ would be well approximated by a constant on $D_P$. While a constant function is easily learnable and may appear to yield a low test loss on $p(x)$, it grossly differs from the true target on $q(x)$. Thus, in terms of generalization, the bound is useful for gauging the generalization properties on the analytically tractable measure $q(x)$, rather than on the empirical measure $(p(x))$ on which the GP would seem to perform very well. In scenarios where a complex feature (such as parity) on the ideal measure $q(x)$ is well approximated by much simpler ones (e.g. a constant) on the training measure one should be wary of associating an unlearnable target, in the sense of our bound, with poor generalization performance within the training distribution.

*Measuring learnability on $q(x)$.* As the example above manifests, it is crucial to note our bound essentially bounds the learnability from the perspective of $q(x)$ as can be seen in eq.(6). That means even for $P < P^*$ the model can perform well on the training dataset and even test datasets. Prominently, this can happen when training (resp. testing) on a low-entropy distribution that can collapse complicated functions onto simpler ones. In this sense, symmetric measures can be seen as maximal entropy distributions under certain constraints; they thus suggest themselves as a ground upon which OOD generalization can be predicted. For more complicated domains it remains an open question which dataset "truly" reflects the feature $\phi_t(x)$ though we expect these differences to be small in practice.

### 4.3 Learning Copying Heads with Transformer

In this example, we consider a transformer with the task of copying the previous token for each token in the sequence. This is an example of a simple closed-form target function that has been shown empirically to have real-world importance. The formation of copying heads is essential for the creation of induction heads and mesa-optimization algorithms (Olsson et al., 2022; von Oswald et al., 2023), which are fundamental in-context learning (ICL) mechanisms. Thus, this simple example can be used as a lower bound on sample complexity for a wide variety of ICL mechanisms.

We define the input as $[X]_i^a$ where $i = 1, ..., V+1$ in the vocabulary index, and $a = 1, ..., L+1$ is the position of the token in the sequence. In this notation the target is $[Y(X)]_i^a = [X]_i^{a-1}$. For simplicity, we introduce a vector notation where $\vec{x}^a$ is a slice of $X$ at the position $a$, and a scalar notation $x_i^a = [X]_i^a$. We note that we do not consider causal masking in this example, as the target function, as defined in the simplified setup above, does not rely on a notion of causality.

As usual, we choose a simple symmetric distribution $q(X)$ where all tokens within sequences and samples are drawn uniformly i.i.d. from the vocabulary and are one-hot encoded, i.e. integers $v = 1, ..., V+1$ such that $x_i^a = \delta_{i,v}$ where $\delta$ is the Kronecker delta. The training dataset $D_P$ will be one-hot encoded, with the same vocabulary size $V+1$, but with arbitrary dependencies between the tokens. In particular, $D_P$ can be a true natural language dataset, tokenized with vocabulary size $V+1$.

The target function $Y(X)$ can be shown to include the feature (see Appendix E)

$$\vec{\phi}_t^a(X) = \frac{1}{z}\left(\vec{x}^{a-1} - \frac{1}{L}\sum_{b=1}^{L}\vec{x}^b - \frac{1}{V}\right); \quad z = \sqrt{L\left(1 - L^{-1} + L^{-1}V^{-1}\right)}, \tag{23}$$

with an eigenvalue that can be bounded from above $\lambda_t \leq \frac{\mathbb{E}_{X \sim q}[k(X,X)]}{(L-1)(V-1)}$. We next use this result together with our main result (8) to bound the sample complexity of copying heads.

Under the assumption of one-hot encoded input $\mathbb{E}_{X \sim D_P}\left[\text{Tr}[Y(X)Y^T(X)]\right] = \mathbb{E}_{X \sim D_P}\left[\sum_{a=1}^{L} \vec{y}^a(X) \cdot \vec{y}^a(X)\right] = L$ for all $D_P$. Finally

$$
\begin{aligned}
\mathbb{E}_{X \sim D_P}\left[\text{Tr}[\Phi(X)\Phi^T(X)]\right] &= \mathbb{E}_{X \sim D_P}\left[\sum_{a=1}^{L} \vec{\phi}_t^a(X) \cdot \vec{\phi}_t^a(X)\right] \\
&= z^{-2}L\left(1 - L^{-2}\sum_{a,b=1}^{L}\sum_{i=1}^{V}\mathbb{E}_{X \sim D_P}[x_i^a x_i^b] + V^{-1}\right)
\end{aligned}
\tag{24}
$$

depends on the choice of $D_P$, nevertheless, one can easily derive bounds on the quantity. A simple bound is $\mathbb{E}_{X \sim D_P}\left[\text{Tr}[\Phi(X)\Phi^T(X)]\right] \leq z^{-2}L(1 - L^{-1}) + V^{-1}$ for any one-hot encoded input.

These results can be plugged in to (8) to find a general lower bound on the sample complexity

$$
P^* \geq \sigma^2 \lambda_t^{-1}(1-\epsilon)\frac{z}{\sqrt{\mathbb{E}_{X \sim D_P}\left[\text{Tr}[\Phi(X)\Phi^T(X)]\right]}} \geq \sigma^2(1-\epsilon)\frac{(L-1)(V-1)}{\mathbb{E}_{X \sim q}[k(X,X)]}\frac{1 - L^{-1} + L^{-1}V^{-1}}{\sqrt{1 - L^{-1} + V^{-1}}}.
\tag{25}
$$

Assuming a long context $L \gg 1$, a large vocabulary $V \gg 1$ and normalized kernel $\mathbb{E}_{X \sim q}[k(X,X)] \simeq 1$, the sample complexity bound simply reads

$$
P^* \gtrsim \sigma^2(1-\epsilon)LV.
\tag{26}
$$

In simple terms, the number of samples has to scale like the product of the context length and the vocabulary size to learn copying heads. This result can be seen as a limit to when models in the kernel limit can start performing in-context learning. We note Olsson et al. (2022) used $V = 2^{16}$ and $L = 8192$ (giving $L \cdot V \approx 0.5 \cdot 10^9$) and found induction heads appear after training on around $2 \cdot 10^9$ tokens for a wide range of model sizes.

## 5 OUTLOOK

In this work, we have shown it is possible to lower bound the sample complexity for kernel ridge regression and Gaussian process regression on a general training dataset based on its symmetric counterpart. Our bound allows exploiting the full symmetries of the kernel, even ones that are not present in the training dataset and thus, naively, cannot be used. The enhanced symmetry of an eigenvalue problem on an auxiliary dataset that possesses all the symmetries of the kernel can greatly simplify its solution.

Kernel ridge regression and Gaussian process regression are not only well-motivated and studied frameworks but are also used in the study of neural networks through the NTK and NNGP correspondences (Jacot et al., 2018; Lee et al., 2018). These correspondences have been used to characterize the inductive bias of different architectures such as fully connected networks (Basri et al., 2019; Yang & Salman, 2020; Scetbon & Harchaoui, 2021; Bietti & Bach, 2021), CNNs (Bietti, 2021; Xiao, 2022; Cagnetta et al., 2023) and transformers (Lavie et al., 2024) under symmetry assumptions for the dataset. Our work allows to generalize these results to datasets that lack these symmetries. We further presented three examples for real-life usage of the bound: (1) a simple application of the bound for linear regression, (2) the hardness of learning parity on a general (correlated) dataset with an FCN, and (3) an application of the bound to copying heads, a stepping stone for ICL.

Much of the possibilities opened by our approach remain unexplored, like tightly bounding the sample complexity in cases where the target is very multi-spectral, which we elaborate on in Appendix C. Another intriguing aspect is the application of such bound to ridgeless regression. When analyzing the true dataset $D_P$, it has been shown there is an effective ridge coming from the unlearnable features (Cohen et al., 2021; Canatar et al., 2021b; Simon et al., 2023). It would be of great interest to find a similar effective ridge for our setting, using the eigenvalues and vectors from the symmetric distribution $q(x)$, without having to solve the eigenvalue problem on $D_P$. Finally, while our bound can be used to characterize what targets will not be learnable, a guarantee on the tightness of the bound can extend its applicability even further.

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

## A  PROOF OF THE MAIN THEOREM

The main idea of the proof builds upon the fact that while the eigendecomposition of the operator

$$\hat{K}_q \phi_i(x) = \int k(x, x') \phi_i(x') q(x') dx' = \lambda_i \phi_i(x) \tag{27}$$

depends on the measure $q(x)$, the representation of the kernel found by Mercer decomposition holds for all datasets within the support on $q(x)$. We then use a known Mercer decomposition on $q(x)$ to represent the kernel function that is sampled in the predictor $\hat{f}_D$ from $D_P$.

The usual GPR/KRR predictor on the dataset $D_P$ is given by Eq.(1). Using Mercer's theorem (König, 1986) to decompose the kernel function to eigenfunctions *on* $q(x)$, the predictor can be written as

$$\hat{f}_D(x) = \sum_{\nu\rho=1}^{P} \sum_{i=1}^{\infty} \phi_i(x) \lambda_i \phi_i(x_\nu) \left[K + I\sigma^2\right]_{\nu\rho}^{-1} y(x_\rho); \quad \hat{K}_q \phi_i(x) = \lambda_i \phi_i(x). \tag{28}$$

We now project the predictor onto the target feature $\phi_t$ with the inner product $\langle \cdot, \cdot \rangle_q$ defined with $q(x)$ as a weighting function

$$\left\langle \phi_t, \hat{f}_D \right\rangle_q = \sum_{\nu\rho} \sum_{i=1}^{\infty} \langle \phi_t, \phi_i \rangle_q \lambda_i \phi_i(x_\nu) \left[K + I\sigma^2\right]_{\nu\rho}^{-1} y(x_\rho) = \lambda_t \sum_{\nu\rho} \phi_t(x_\nu) \left[K + I\sigma^2\right]_{\nu\rho}^{-1} y(x_\rho), \tag{29}$$

where the inner product can be carried out immediately based on the orthonormality of $\{\phi_i\}_i$ w.r.t. the inner product $\langle \cdot, \cdot \rangle_q$. We may now use Cauchy-Schwartz inequality to bound the inner product on $D_P$, given by the summation on the index $\nu$,

$$\left| \left\langle \phi_t, \hat{f}_D \right\rangle_q \right| \leq \lambda_t \sqrt{\left(\sum_\nu \phi_t(x_\nu) \phi_t(x_\nu)\right) \sigma^{-4} \left(\sum_{\mu\rho\sigma} [\sigma^{-2} K + I]_{\mu\rho}^{-1} y_\rho [\sigma^{-2} K + I]_{\mu\sigma}^{-1} y_\sigma\right)}. \tag{30}$$

Lastly, since $\left[\left(\sigma^{-2} K + I\right)\right]^{-1}$ is weakly contracting we can bound the result from above by

$$\left| \left\langle \phi_t, \hat{f}_D \right\rangle_q \right| \leq \sigma^{-2} \lambda_t \sqrt{\left(\sum_\nu \phi_t(x_\nu) \phi_t(x_\nu)\right) \left(\sum_\mu y_\mu y_\mu\right)} = \sigma^{-2} \lambda_t P \sqrt{\mathbb{E}_{x\sim D_P}\left[\phi_t^2(x)\right] \mathbb{E}_{x\sim D_P}\left[y^2(x)\right]}. \tag{31}$$

To quantify whether the feature is learned or not we may compare the projection of the predictor on $\phi_t$ to that of the target function and require their ratio to be $1 - \epsilon$, i.e. require that the learnability is $\epsilon$ close to perfect

$$\left| \frac{\left\langle \phi_t, \hat{f}_D \right\rangle_q}{\langle \phi_t, y \rangle_q} \right| \overset{!}{=} 1 - \epsilon, \tag{32}$$

yielding the result in Eq.8.

## B  FURTHER RELATED WORKS

*Symmetry in neural networks & sample complexity.* Symmetry has been used extensively to understand neural networks, especially from the perspective of sample complexity and the kernel regime. Fully connected networks with data uniformly distributed on a hypersphere have been studied in Basri et al. (2019); Bietti & Bach (2021); Scetbon & Harchaoui (2021). Data distributed non-uniformly on the hypersphere was studied in Basri et al. (2020), while Gaussian data and data uniformly distributed on the hypercube were studied in Yang & Salman (2020). This line of study was extended beyond the natural rotation symmetry, Bietti et al. (2021) studied how extending the symmetry group beyond the rotational symmetry of fully connected networks reduces the sample complexity a result that has been generalized further by Tahmasebi & Jegelka (2023). The eigenspectra of kernels corresponding to convolutional neural networks were studied in Bietti (2021); Xiao (2022); Cagnetta et al. (2023);

Geifman et al. (2022). Finally, Lavie et al. (2024) studied transformers in the kernel regime by leveraging their permutation symmetry. As a future research direction, it can interesting to try and use this rich literature on distributional shift for the purpose proposed in this paper.

*Learning & multiple data distributions.* There is a large body of work on distributional shift and out-of-distribution generalization Pan & Yang (2010); Ben-David et al. (2010); Sugiyama & Kawanabe (2012); Arjovsky (2021); Canatar et al. (2021a); Ma et al. (2023), however, this setting and its motivation are different from ours. In the study of distributional shifts, one assumes the training distribution does not reflect accurately the test distribution and tries to estimate (bound, guarantee) the performance on the test distribution, which is the true object of interest. This concept fundamentally differs from the setting in this work. Here we do not assume any difference between test and train distribution, rather, we are interested in the performance on the dataset $D_P$ and use $q$, the auxiliary measure, solely as a tool. Presenting $q$ allows us to capitalize on all the results mentioned in the previous paragraph about symmetry, even when the dataset $D_P$ does not respect those symmetries. We are aware of one previous work Opper & Vivarelli (1998) that used a similar motif to upper bound the learnability, however, here we lower bound it. Moreover, Opper & Vivarelli (1998) did not suggest a specific dataset that can be used to simplify the problem, and does not mention symmetries.

## C  MULTI-SPECTRAL EXTENSIONS

In the main text, we bound the magnitude of the predictor's projection onto the kernel's eigenfunctions on $q(x)$ and compare it to those of the target. Here we consider the case where $y(x)$ is highly multispectral and receives contributions from a large, potentially infinite, number of $\phi_t$ modes. In such a scenario, $\langle \phi_t, y \rangle_q^2$ would scale inversely with the number of dominant modes whereas $y^2$ appearing in our bound would remain $O(1)$. As a consequence, the bound may become very loose as is the case where the data distribution and $q(x)$ match, where our bound would be $O\left(\langle \phi_t, y \rangle_q^{-2}\right)$ off the EK result.

Here we extend our bound to certain multispectral circumstances. Specifically, let us assume that $y(x)$ can be written as $y_<(x) + y_>(x)$ such that $y_>(x)$ is spanned by $\{\phi_i(x)\}_i$ having $\lambda_i \leq \lambda_>$. Furthermore let us assume $|y|^2 \propto |y_>(x)|^2 \propto O(1)$ and $|y_>(x)|_q^2 \geq O(1)$. As we argue next in such cases we essentially derive a similar bound with $\lambda_>$ playing the role of $\lambda_t$. Specifically we consider $|\langle \hat{f}_D(x), y_>(x) \rangle|_q$ given by

$$| \int dx q(x) y_>(x) \sum_{\mu\nu} K(x, x_\mu)[K + I\sigma^2]_{\mu\nu}^{-1} y_\nu| = | \sum_{t \geq t_>} \lambda_t \langle \phi_t, y \rangle_q \phi_t(x_\mu)[K + I\sigma^2]_{\mu\nu}^{-1} y_\nu| \quad (33)$$

$$= |[\sum_{t \geq t_>} \lambda_t \langle \phi_t, y \rangle_q \vec{\phi}_t]^T [K + I\sigma^2]^{-1} \vec{y}] \leq \sqrt{||[\sum_{t \geq t_>} \lambda_t \langle \phi_t, y \rangle_q \vec{\phi}_t]||^2 ||[K + I\sigma^2]^{-1} \vec{y}||^2}$$

$$\leq \sigma^{-2} \sqrt{||[\sum_{t \geq t_>} \lambda_t \langle \phi_t, y \rangle_q \vec{\phi}_t]||^2 ||\vec{y}||^2} = \lambda_> \sigma^{-2} \sqrt{\sum_\mu \left[\sum_{t \geq t_>} \frac{\lambda_t}{\lambda_>} \langle \phi_t, y \rangle_q \phi_t(x_\mu)\right]^2 ||\vec{y}||^2}$$

Notably, if all $\lambda_t$'s are degenerate, we retrieve our previous bound with the feature being $\sum_t \langle \phi_t, y \rangle_q \phi_t(x)$. More generally, we need to average the feature $\sum_t \frac{\lambda_t}{\lambda_>} \langle \phi_t, y \rangle_q \phi_t(x)$ squared over the training set.

Here we suggest several ways of treating this latter average. In some cases, where we have good control of all $\lambda_t$'s and $\phi_t$'s, we may know how to bound this last quantity directly. Alternatively, we may write

$$\sum_\mu \left[\sum_{t \geq t_>} \frac{\lambda_t}{\lambda_>} \langle \phi_t, y \rangle_q \phi_t(x_\mu)\right]^2 = \sum_{t,p \geq t_>} \frac{\lambda_t}{\lambda_>} \frac{\lambda_p}{\lambda_>} \langle \phi_t, y \rangle_q \langle \phi_u, y \rangle_q \left[\vec{\phi}_t^T \vec{\phi}_u\right] \quad (34)$$

and argue that in a typical scenario, $\vec{\phi}_t^T \vec{\phi}_u$ with $u \neq t$ would be smaller and also sum up incoherently. The diagonal contributions would thus be the dominant ones. Further assurance may be obtained

by sampling $\lambda_t \lambda_u \langle \phi_t, y \rangle_q \langle \phi_u, y \rangle_q \vec{\phi}_t^T \vec{\phi}_u$ and verifying that off-diagonal contributions are indeed smaller and incoherent. Considering the diagonal contribution alone we obtain

$$|\langle \hat{f}_D(x), y_>(x) \rangle|_q \le P \lambda_> (1 - \epsilon) \sigma^{-2} \sqrt{\sum_{t \ge t_>} \left( \frac{\lambda_t}{\lambda_>} \right)^2 \langle \phi_t, y \rangle_q^2 \, \mathbb{E}_{D_P}[\phi_t^2] \mathbb{E}_{D_P}[y^2]} \tag{35}$$

which recalling $\lambda_t / \lambda_> \le 1$ and $\sum_{t \ge t_>} [\langle \phi_t, y \rangle_q]^2 = O(1)$ yields a similar result to before.

Otherwise, we can take a worst-case scenario in which all $\phi_t(x_\mu)$ contribute coherently to the sum [indeed think of $\phi_t$ as 1d Fourier modes, and the training distribution is a delta function at zero, and we wish to learn a delta function of the training set. Our $[K + \sigma^2]^{-1} y \approx \sigma^{-2} y$ estimate would be very poor however the $\phi_t$'s would all sum coherently around zero]. In this case, we may use Cauchy Schwarz again on the summation over $t$ to obtain

$$|\langle \hat{f}_D(x), y_>(x) \rangle|_q \le \lambda_> \sigma^{-2} (1 - \epsilon) \sqrt{\sum_{t \ge t_>} |\langle \phi_t, y \rangle_q|^2 \sum_{\mu, t} \left| \frac{\lambda_t}{\lambda_>} \phi_t(x_\mu) \right|^2 ||\vec{y}||^2} \tag{36}$$

Consider this as a learnability namely $\langle f, y_> \rangle_q / \langle y_>, y_> \rangle_q$ bearing in mind that $y_>(x)$ has an $O(1)$ norm (or more) also under $q(x)$. Scaling-wise, we may thus remove the $\sum_{t \ge t_>} |\langle \phi_t, y \rangle_q|^2$ factor. Doing so we retrieve our previous bound, only with $\sum_t \frac{\lambda_t^2}{\lambda_>^2} E_D(\phi_t^2)$ instead of just $E_D(\phi_t^2)$.

Notably, since in our normalization all $\lambda_t < 1$, $\sum_t \lambda_t^2$ decays to zero quicker than $\lambda_t$ and hence by the finiteness of the trace yields a finite number even if $y_>$ contains an infinite amount of features.

## D    CALCULATING THE NORMALIZATION FACTOR FOR PARITY & FCN

To calculate this normalization factor we first extend it to be a function of the sphere radius namely

$$\hat{n}[r] = \left( \frac{r^{d-1} 2\pi^{d/2}}{\Gamma(d/2)} \right)^{-1} \int_{R^d} d^d x \, \delta(|x| - r) \prod_{i=1}^d x_i^2 \equiv \left( \frac{r^{d-1} 2\pi^{d/2}}{\Gamma(d/2)} \right)^{-1} n[r], \tag{37}$$

where the first factor is the hypersphere surface area in $d$ dimensions under the assumption of even $d$. While we are interested in $n[r = \sqrt{d}]$ we instead first look at

$$N[s] = \int_0^\infty dr \, e^{-sr^2} n[r] = \int_0^\infty d(r^2) e^{-sr^2} \frac{n[r]}{2r} \tag{38}$$

Notably $N[s]$ is then the Laplace transform of $n[r]/(2r)$ viewed as a function of $r^2$. Calculating it based on the second expression amounts to independent Gaussian integrations and yields

$$N[s] = \int_{R^d} d^d x \, e^{-s|x|^2} \prod_i x_i^2 = \left( \frac{\pi}{2^2 s^3} \right)^{d/2} \tag{39}$$

Inverting this Laplace transform we obtain

$$\frac{n[r]}{2r} = \frac{2^{-d} \pi^{d/2} r^{3d-2}}{\Gamma\left( \frac{3d}{2} \right)} \Rightarrow n[r] = \frac{2^{-(d-1)} \pi^{d/2} r^{3d-1}}{\Gamma\left( \frac{3d}{2} \right)} \tag{40}$$

$$\hat{n}[r] = \left( \frac{r^{d-1} 2\pi^{d/2}}{\Gamma(d/2)} \right)^{-1} n[r] = \frac{2^{-d} r^{2d} \Gamma\left( \frac{d}{2} \right)}{\Gamma\left( \frac{3d}{2} \right)}$$

## E    EIGENVALUES FOR COPYING HEADS

For the data distribution $q(X)$ we may use the results of (Lavie et al., 2024) to characterize the eigenvalues and eigenvectors of the NNGP/NTK kernel of a transformer. The approach relies on the permutation symmetry between tokens in the same sample and uses representation theory to

upper bound eigenvalues based on their degeneracy. Eigenvalues that belong to the same irreducible representation (irrep) $R$ and degenerate subspace $V_R^i$ can be bounded by the kernel's trace over the dimension of the irrep

$$\lambda_{V_R^i} \leq \frac{\mathbb{E}_{x \sim q}[k(x,x)]}{\dim_R}, \tag{41}$$

where $\lambda_{V_R^i}$ is the $\dim_R$-fold degenerate eigenvalue of the subspace $V_R^i$, $E_{x \sim q}[k(x,x)]$ is the kernel's trace and $\dim_R$ is the dimension of the irrep.

In the case of $q(x)$ from the main text, we have a permutation symmetry in sequence space and an additional permutation symmetry in vocabulary space, since it is uniformly distributed. We can use this additional symmetry to identify the spaces $V_R^i$ mentioned above within the space of the irrep $R$ (i.e. within the space that includes the multiplicity of $R$).

We thus set to decompose the target function into irreps of the symmetric group. As shown in (Lavie et al., 2024) linear functions (such as a copying head) are decomposed into two irreps of the symmetric group, "trivial" and "standard". We will look and the "standard" component both in sequence space and in vocabulary space in order to capture the most unlearnable feature that is required for the copy head target.

$$\vec{\phi}_t^a(X) = \frac{1}{z}\left(\vec{x}^{a-1} - \frac{1}{L}\sum_{b=1}^{L}\vec{x}^b - \frac{1}{V}\right); \quad z = \sqrt{L}\sqrt{1 - L^{-1} + L^{-1}V^{-1}} = \mathbb{E}_{X \sim q}[\vec{\phi}_t^a(X)\vec{y}^a(X)] \tag{42}$$

It is part of a $(L-1)(V-1)$ degenerate space of the standard irrep of both the vocabulary and sequence permutation symmetry

$$\dim_t = (L-1)(V-1) \tag{43}$$

hence the eigenvalues can be bounded by

$$\lambda_t \leq \frac{\mathbb{E}_{x \sim q}[k(x,x)]}{(L-1)(V-1)}. \tag{44}$$

## F  FURTHER EXPERIMENTAL RESULTS

In Figure 3, left panel, we show an experiment of exact KRR with a kernel that corresponds to a 2-layer ReLU network learning a random quadratic function

$$y(x) = a + \sum_{i=1}^{d} b_i x_i + \sum_{i,j=1}^{d} c_{ij} x_i x_j; \quad a, b_i, c_{ij} \overset{\text{i.i.d.}}{\sim} \mathcal{N}(0,1) \tag{45}$$

where $D_p$ is a dataset of $10^4$ samples drawn uniform i.i.d. on the hypersphere $\mathbb{S}^9$ ($d = 10$). The symmetric auxiliary measure $q(x)$ is naturally chosen to be the underlying symmetric distribution, (continuous) uniform on the hypersphere. Finally, the features $\phi_1, \phi_2$ are chosen to be the projections of the target to the $l = 1, 2$ spaces of the hyperspherical harmonics (linear and quadratic features, respectively). We plot the trainset size $P$ on the horizontal axis and the ratio of the learnability to our bound on it (the l.h.s. of Eq. (7) divided by the r.h.s.)

$$\mathcal{L}_t^{D,q}/B = \frac{\left|\left\langle \phi_t, \hat{f}_D \right\rangle_q\right|}{\sigma^{-1}P^*\lambda_t\sqrt{\mathbb{E}_{x \sim D_P}[\phi_t^2(x)]\,\mathbb{E}_{x \sim D_P}[y^2(x)]}} \tag{46}$$

on the vertical axis. In this case, the bound is seen to approximate the beginning of the learning stage well. The regime in which the bound is tight indicates an important feature of our result. It captures the onset of learning well and thus can be used to judge sample complexity, but misses the saturation of the learnability at later stages of learning. We stress that the dots indicate a single random realization of the dataset, and the bound is guaranteed to hold for every such realization.

Figure 3, right panel, shows the same 2-layer ReLU network KRR learning to perform binary classification of two classes from CIFAR-10. The input dimension is reduced by PCA to $d = 10$. Afterward, each sample is normalized such that $\forall x \in D_p ||x||_2^2 = 1$. Since this scenario uses a

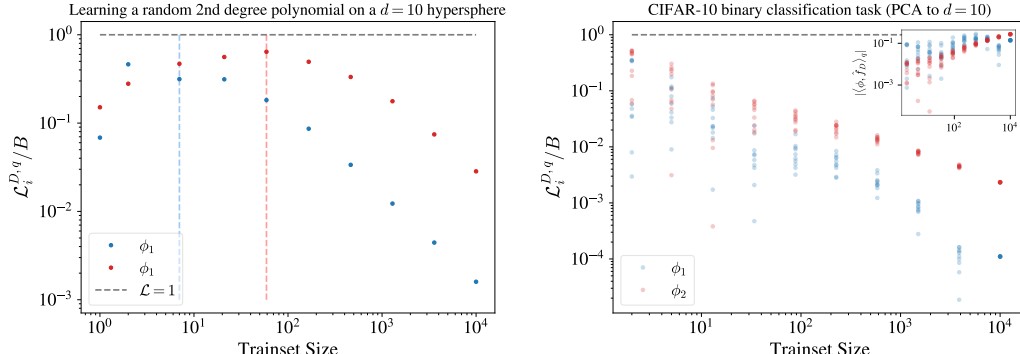

Figure 3: **Left: (Learnability to bound ratio as a function of dataset size)** The learnability computed empirically divided by our learnability bound of a random quadratic target function for linear $\phi_1$ and quadratic $\phi_2$ features. The trainset consists of $10^4$ samples drawn uniformly on the hypersphere $\mathbb{S}^9$ and $q(x)$ is a uniform (continuous) distribution on the hypersphere. The horizontal line indicates a ratio of 1 (perfectly tight bound). The vertical lines indicate the onset of learning defined as the point closest to $\mathcal{L} = 1 - \epsilon = 0.1$. The bound is seen to be tight before and around the onset of learning even for a single realization. Notably, we do not expect the bound to be tight when the feature is already learned well but to predict the minimum required number of samples for learning. **Right: (The bound deployed on classification task for CIFAR-10)** the ratio between the learnability on CIFAR-10 $\mathcal{L}^{\mathrm{CIFAR}-10,q}$ and the learnability bound. $q$ is again chosen to be a uniform (continuous) distribution on the hypersphere, with $\phi_1, \phi_2$ linear and quadratic features respectively. 10 independent realizations are shown. The inset shows the numerator of the cross-dataset learnability and can be used to judge the learnability trend and regime.

real-life empirical task, the extension of the target function to the entire hypersphere is undefined[9]; yet the ratio of the learnability to our bound ((46)) still is. For the features $\phi_t$'s, we again choose the linear and quadratic features that result from projecting the target onto the $l = 1, 2$ spaces of the hyperspherical harmonics (for implementation details see subsection F.1 below). On this plot, our bound again appears as the constant $f(P) = 1$ and predicts that all points will not be above it. As before our bound can be seen to be tight at the beginning of the learning process, capturing the large fluctuations due to larger variability between the dataset realization[10].

The left and right panels of Figure 4 correspond to the same experiments as Figure 3 right panel, performed on binary classification between two classes of MNIST digits, and Fashion MNIST respectively.

## F.1 Further Details

To experimentally compute the mean over a continuous and uniform distribution on the hypersphere, we sampled $10^6$ points uniformly i.i.d. from the hypersphere and computed the mean on these points. Note that this is a much larger number of points than the one used for the trainset $10^4$.

For experiments with real-life datasets $D_p$ (CIFAR-10, MNIST, Fashion MNIST), we extracted the features $phi_1, \phi_2$ by training a 2nd degree surrogate polynomial kernel

$$k(x, y) = \frac{1 + x \cdot y + (x \cdot y)^2}{3} \tag{47}$$

on the dataset $D_p$ resulting in the predictor

$$\hat{f}(x_*) = \sum \mu, \nu = 1^P K_{*,\mu}[K + I\epsilon]^{-1}_{\mu,\nu} y_\nu, \quad \epsilon = 10^{-3}. \tag{48}$$

---

[9]The quantity $\left|\langle \phi_t, y \rangle_q\right|$ is essentially unknowable as one would have to extend the classification of CIFAR-10 to the entire hypersphere, including random-noise images.

[10]for $P = 10^4$ there is a single unique realization - the whole dataset

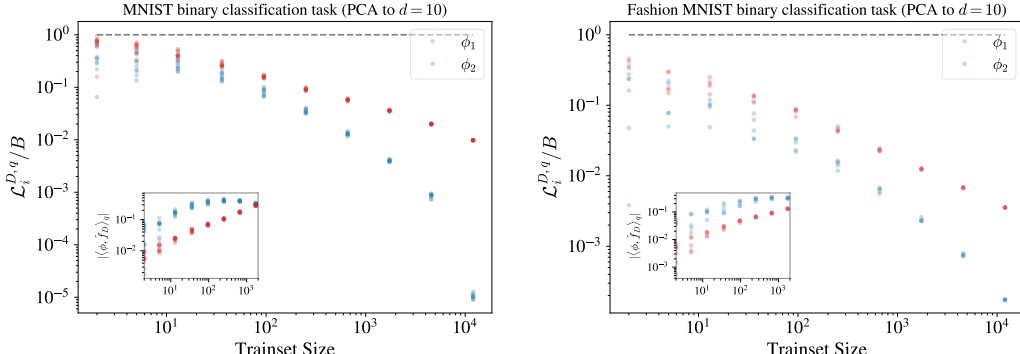

Figure 4: **Left: (Bound for binary classification of MNIST)** the ratio between the learnability on MNIST $\mathcal{L}^{\mathrm{MNIST},q}$ and the learnability bound. $q$ is again chosen as a uniform (continuous) distribution on the hypersphere, with $\phi_1, \phi_2$ linear and quadratic features respectively. The onset of learning is tightly captured by our bound. 10 independent realizations are shown. The inset shows the numerator of the cross-dataset learnability and can be used to judge the learnability trend and regime. **Right: (Bound for binary classification of Fashion MNIST)** Similar to the figure on the left, for classification of Fashion MNIST. 5 independent realizations are shown.

We then extracted the features $\phi_1, \phi_2$ that correspond to hyperspherical harmonics with $l = 1, 2$ respectively by

$$\phi_1(x) := \frac{\hat{f}(x) - \hat{f}(-x)}{\sqrt{\int_{\mathbb{S}^9} (\hat{f}(x) - \hat{f}(-x))^2 dx}} \tag{49}$$

$$\phi_2(x) := \frac{\hat{f}(x) + \hat{f}(-x) - \int_{\mathbb{S}^9} \hat{f}(x) dx}{\sqrt{\int_{\mathbb{S}^9} (\hat{f}(x) + \hat{f}(-x) - \int_{\mathbb{S}^9} \hat{f}(x))^2 dx}}. \tag{50}$$

The integration over the hypersphere is again done numerically as described in the paragraph above.

