# OpenReview forum: "Symmetric Kernels with Non-Symmetric Data: A Data-Agnostic Learnability Bound"
_ICLR.cc/2025/Conference — Submitted to ICLR 2025_

### Official Review · Reviewer_LiQC · 2024-10-25

**Soundness:** 3
**Presentation:** 2
**Contribution:** 2
**Rating:** 6
**Confidence:** 2

**Summary:**

The paper presents novel techniques to analyse the learnability of kernel methods such as kernel ridge regression and Gaussian process regression, including on real-world datasets that exhibit non-symmetric distributions. Traditional definitions of learnability require diagonalising the kernel on the data measure, but this is computationally expensive and requires access to the underlying data measure. The authors propose to mitigate this by defining a new notion of cross-dataset learnability, where they learn on the empirical data measure $D_p$ and judge functional similarity (‘perform evaluation’) on an auxiliary measure $q(x)$. This auxiliary measure is chosen judiciously to have particular symmetry properties, which makes bounding its eigenvalues tractable: one decomposes the target function into irreducible representations of the symmetry group and uses their inverse dimension to bound the corresponding eigenvalue. Thm. 3.1 gives an upper bound for sample complexity for this new notion of cross-dataset learnability. The authors demonstrate on real world and synthetic data, and provide three ‘vignettes’ to demonstrate their workflow and the types of conclusions one can draw using their approach.

**Strengths:**

The manuscript makes strong technical contributions. I can see how introducing cross-dataset learnability opens up exploiting the symmetry properties of an idealised auxiliary measure using tools for representation theory, including in situations where this symmetry is not respected in the true real-world dataset. It provides a natural avenue to port over existing results for idealised, symmetric datasets (Yang and Salman (2021), Lavie et al. (2024), etc.). The authors work hard to provide intuition for the reader, and the ‘vignettes’ showcase a range of interesting applications. The experiments include results for synthetic and real-world data.

**Weaknesses:**

1. *Articulation of the relationship between conventional and cross-dataset learnability*. I understand that the ability to diagonalise a simplified, symmetric kernel rather than its more challenging counterpart on the empirical dataset measure or true underlying dataset measure brings strong computational benefits. But can the authors say anything more concrete about the relationship, mathematically or even heuristically, between these different notions of learnability? Clearly they agree when learning from $p(x)$ (rather than $D_p$) and when $q(x)=p(x)$ (line 147), but can anything be added about e.g. small departures from this regime? ‘Comparison with other measures of learnability’ (line 421) discusses how the result in Sec. 4.2. can be interpreted as an ‘unlearnability bound’ which agrees heuristically with existing results in the literature. This type of observation helps convince the reader that cross-dataset learnabilty is practically useful; I think the manuscript would benefit from further similar discussion. Supplementing Sec. 2 to more carefully relate Eq. 4 and Eq. 6 could also help.
2. *Utility on real-world data and Figures 1 and 2*. In line 244, the authors note that their result ‘misses the saturation of learnability at later stages of learning’. I think this refers to how learnability drops as the trainset size grows. Presumably this is because the measures on which we learn and judge functional similarity now differ; we measure overlap of the predictor with eigenfunctions $\phi_t$ of the kernel wrt $q(x)$, but are training using $D_p$. If I understand this correctly, it seems that cross-dataset learnability decreases monotonically for trainset size for CIFAR-10, MNIST and Fasion-MNIST. Of course, there is a tradeoff in choosing an auxiliary measure $q(x)$ that is easier to diagonalise (to derive the bound) but also reflects the true data distribution less well, but isn’t it a problem that this formulation doesn’t appear to capture conventional notions of learnability at all for all three real world datasets? I’m also a bit worried about the comment that ‘the input dimension is reduced by PCA to $d=10$ as a balance between the dimension and the number of samples that was empirically found to allow learning of quadratic features’ (line 250): how robust are these experimental results? As a more stylistic comment, I can’t see any reference to the inset panes of Figs 1 and 2; it would be great to add discussion of what these mean.
3. *Choosing $q(x)$*. To add to the above: the authors note that one is free to choose the most favourable $q(x)$ in the class satisfying Eq. 8. I wonder if they can expand on this important point, discussing the tension between the presence of simplifying symmetries and how meaningful cross–dataset learnability is in practice. Can they provide any heuristic guidance for practitioners? More stylistically, I think it would be really helpful to see a table or schematic listing existing results for symmetry groups, their corresponding architectures, eigenvalue bounds and references.
4. *Transformer results*. As early as line 21 of the abstract the authors highlight their results for sample complexity of copying heads for kernels associated with transformers. But in the main text and appendix these sections are terse. How is one to interpret the sample complexity bound in Eq. 26? I’d suggest either adding extra discussion here, or shifting focus in the abstract to highlight your very nice broader experimental contributions.
5. *Presentation and minor comments*.
- I found the paragraph beginning ‘taking a Bayesian viewpoint’ (line 63) difficult to parse and somewhat disconnected from the rest of the introduction. I think the authors want to argue that expecting exact symmetry in the true data distribution is unrealistic and motivate instead using auxiliary $q(x)$ through a Bayesian lens, but I wonder if this could be rephrased a little more clearly.
- As minor typographical points, line 89 is missing a closing parenthesis and line 413 references an equation in the Appendix (presumably because of degenerate labelling). I would also consider splitting Eq. 25 onto multiple lines.

**Questions:**

1. Can the authors comment more on the relationship between conventional measures of learnability and their new scheme?
2. Can the authors provide interpretation of the sample complexity bound in Eq. 26, especially the dependence on $V$ and $L$?
3. Can the authors clarify the discussion about the rotational symmetry of the FCN on line 54? The references aren’t totally clear to me. Perhaps they refer to Thm 3.1 of Basri (2019), but I think this would benefit from further details and a more careful discussion of the necessary assumptions.

I thank the authors for their efforts and nice submission. If they can explain the relationship to conventional learnability and the practical implications of their scheme (or convince me why these things shouldn’t matter), I will be happy to raise my score.

---

> ### Author Response · Authors · 2024-11-20
>
> We thank the reviewer for the in-depth review of our work and the insightful comments. We have made several changes to the manuscript based on these and we believe they have greatly improved the paper. Below we respond to the specific questions and comments.
> - *Articulation of the relationship between conventional and cross-dataset learnability*.
>
> **Response**: We thank the reviewer for drawing our attention to this issue. We have made the similarity between the definitions of the two quantities more explicit, added a fuller discussion to section 2, and moved the discussion from section 4.2 up to section 2. Moreover, we have performed new experiments and now present new results in the main text, we believe these help address this issue. Specifically, Figure 1 now shows the “ordinary” learnability together with the cross-dataset learnability to make the relation between the two more tangible; Figure 2 now shows results for whitened data which has a spherically symmetric covariance thereby bringing it closer to $q$, and compares these results with non whitened data.
> - *Utility on real-world data and Figures 1 and 2.*
>
> **Response**: We apologize for this, it seems the figures failed to communicate the results. First, we clarify some of the points raised by the reviewer
> - Note that what is shown in those figures is the ratio between the cross-dataset learnability and our bound. The drop in those figures is not the result of the cross-dataset learnability decreasing but rather a result of it reaching a constant value (saturating) while the bound continues to grow linearly with P indefinitely.
>  - We delegated these figures to an appendix and now use figures that show the common learnability alongside the cross-dataset learnability and indicate the bound on the number of samples required for learning.
>
> In the new figures, the cross-dataset learnability is seen to be indicative of the on-dataset learnability, and the sample complexity bound is shown to be practical.
>
> Reducing the dimension was merely a numerical convenience, we wanted to work with a not-too-large dataset of $~10000$ samples (2 classes of CIFAR/MNIST/F-MNIST) but still be able to have non-negligible cross-dataset learnability. Since the eigenvalue associated is a quadratic feature of a rotationally invariant kernel would scale as $d^{-2}$ taking, for example, $d=100$ would already give $P^* \approx 10000$, forcing us to use a large dataset. The new experiments use larger dimension ($d=14$ for Fig.1 and $d=20$ for Fig. 2) these dimensions match the effective dimension of these datasets as was measured empirically [1-4]
>
> - *Choosing $q(x)$*.
>
> **Response**: This is an interesting point, we thank the reviewer for bringing it up. While giving an exact answer to this question lies beyond the scope of the current work, we did perform new experiments to address it. In the new experiments, shown in the new Fig.2, we use PCA whitening rather than regular PCA, resulting in a covariance matrix that is spherically symmetric. The results are indeed closer together in this case.
>
> Additionally, we can certainly provide guidelines on choosing $q(x)$. One wants to balance maximizing the symmetries and choosing a distribution similar to the true one. On the one hand, larger symmetry gives a tighter bound on the eigenvalues and more freedom in choosing a relevant target feature within each irrep (as the irreps are larger). On the other hand, one does not want to create a distribution that would make the cross-dataset learnability very different from the on-dataset learnability. As shown in the new experiments, there is quite a large range of choices that will yield useful results; e.g. Fashion MNIST images are certainly not uniformly sampled from the hypersphere yet we get good sample complexity results for this $q$.
>
> Finally, We thank the reviewer for the advice on the table of symmetries, architectures, and eigenvalues, while not present yet in the current version we will add one to the final version. Such a table could include FCNs and their rotational symmetry, transformers and their permutation symmetry, as well as architectures designed to respect specific symmetries like G-CNNs [5]. We hope this work will motivate even further work along this line.
>
> [1] Pope P, Zhu C, Abdelkader A, Goldblum M and Goldstein T 2021 THE INTRINSIC DIMENSION OF IMAGES AND ITS IMPACT ON LEARNING
>
> [2] Aumüller M and Ceccarello M 2021 The role of local dimensionality measures in benchmarking nearest neighbor search
>
> [3] Facco E, d’Errico M, Rodriguez A and Laio A 2017 Estimating the intrinsic dimension of datasets by a minimal neighborhood
>
> [4] Spigler S, Geiger M and Wyart M 2020 Asymptotic learning curves of kernel methods: empirical data v.s. Teacher-Student paradigm
>
> [5] Cohen T S and Welling M 2016 Group Equivariant Convolutional Networks

---

> ### Author Response · Authors · 2024-11-20
>
> - *Transformer results.*
>
> **Response**: We thank the reviewer for shedding light on this issue. We now further explain the result and also give a back-of-the-envelope calculation that shows the appearance of induction heads in large-scale experiments much the scaling given by our bound. We cite here the new paragraph:
> >“Assuming a long context $L \gg 1$, a large vocabulary $V \gg 1$ and normalized kernel $\mathbb{E}_{X \sim q}[k(X,X)] \simeq 1$, the sample complexity bound simply reads
> \begin{equation}
> 	P^* \gtrsim \sigma^2 (1-\epsilon) L V.
> \end{equation}
> In simple terms, the number of samples has to scale like the product of the context length and the vocabulary size to learn copying heads. This result can be seen as a limit to when models in the kernel limit can start performing in-context learning. We note (Olsson et al. 2022) used $V=2^{16}$ and $L=8192$ (giving $L\cdot V\approx0.5\cdot10^9$) and found induction heads appear after training on around $2 \cdot 10^9$ tokens for a wide range of model sizes.”
> - *I found the paragraph beginning ‘taking a Bayesian viewpoint’ (line 63) difficult to parse...*
>
> **Response**: We greatly appreciate the attention to the manuscript and this valuable comment. We agree that this paragraph was a weak point in the introduction and have changed it in the updated version.
> - *As minor typographical points…*
>
> **Response**: We are grateful for the reviewer's help in improving the manuscript! We fixed these in the new version.
> - *Can the authors comment more on the relationship between conventional measures of learnability and their new scheme?*
>
> **Response**: We thank the reviewer for highlighting the question, we now address it more fully in the text and in the experiments. This part is given below
> >“Comparing equations (4), (6) the inner product (equivalently expectation) in the cross-data learnability is taken over the auxiliary measure instead of on the dataset. This change can be intuitively understood as learning from $D_P$, but judging how good the reconstruction is based on functional similarity on $q(x)$. Such a change can have the advantage of weighting all possible inputs evenly, perhaps capturing a notion of out-of-distribution generalization but the disadvantage of being uninformed about the details of the specific dataset. The issue of judging similarity differently based on different measures is exemplified and discussed further in Section 4.2.
> > Cross-dataset learnability agrees with the common one when learning from the true underlying probability density of the dataset $p(x)$ (rather than the empirical $D_P$) and choosing $q(x) = p(x)$. However, The underlying density $p(x)$ is often inaccessible, in these cases, one can still use our generalized learnability to estimate learnability from the empirical dataset $D_P$.
> > We note that while the on-dataset learnability in Eq.(4) is bounded $\mathcal{L}_t \in [0,1)$, the cross-dataset learnability is unbounded from above. As a consequence, maximizing cross-dataset learnability does not imply good learning; instead, one must require it to be close to unity.”
>
> The experiments now compare the two kinds of learnability and show sample complexity derived by cross-dataset learnability is relevant for on-dataset learnability. This relationship is even strengthened when the distributions are more similar as the experiments with whitened data show.
>
> - *Can the authors provide interpretation of the sample complexity bound in Eq. 26, especially the dependence $L$ on and $V$?*
>
> **Response**: This can be seen as a case of the curse of dimensionality: larger context and vocabulary make it harder to learn and require more samples. We note that this is also seen in a range of real-world large transformers outside the kernel regime. Olsson et al. 2022 consistently found it takes more than a billion tokens before induction heads are learned, this is a non-trivial amount. A billion tokens are approximately 10 times more than the large version of WikiText or roughly 1% of the gigantic C4 dataset.
>
> - *Can the authors clarify the discussion about the rotational symmetry of the FCN on line 54?*
>
> **Response**: These references may indeed not be the most approachable, we cite more approachable ones in section 3.1. We now cite them on line 54 as well. To give the (approximate) gist of these results:
>
> Fully connected networks result in rotationally symmetric kernels. Rotationally symmetric kernels with rotationally symmetric distributions are diagonalized in the basis of the hyperspherical harmonics. The hyperspherical harmonics are arranged in irreps labeled by $\ell$ which corresponds to their degree. Since the dimension of the $\ell$’th irrep scales as $d^\ell$ the associated eigenvalue scales as $\lambda_\ell \sim d^{-\ell}$.
>
> The works cited in the manuscript give a much more fine-grained understanding of the spectrum, including results that account for the depth of the network, the specific activation functions used, etc.

---

> ### Comment · Reviewer_LiQC · 2024-11-22
>
> Thanks for the response and the comprehensive answers to my questions. I closed my review by saying 'If they can explain the relationship to conventional learnability and the practical implications of their scheme (or convince me why these things shouldn’t matter), I will be happy to raise my score.'
>
> 1. *Relationship to conventional learnability*. I think the manuscript has improved on this point -- Fig. 1 now shows the relationship between learnability computed empirically and cross-data learnability, along with the authors' bounds. This helps convince the reader these things are at least somewhat related. Fig. 2 captures how whitened PCA makes the bound tighter, which heuristically makes sense because the dataset and auxiliary measure become more similar. I think the manuscript would benefit from more discussion like this: perhaps an experiment showing how the strength of correlation between conventional and cross-dataset learnability and bound tightness change as the dataset and auxiliary measure become more similar in a smooth way (beyond either whitened vs regular PCA), or better **some theoretical results relating the two beyond agreement when learning from $p(x)$ (rather than $D_p$) and when $q(x)=p(x)$**.
> 2. *Practical implications of the scheme*. The manuscript has also improved on this point. Thanks for adding extra interpretation of the Transformer bound (now Eq. 25). Reading the responses to other reviewers and the revised text, I can see that the goal is to find a necessary condition for learnability, and that the authors note that the bound to the learnability is always within an order of magnitude of the true learnability during the learning phase. I think this (line 246) could still be spelled out a little more explicitly.
>
> Whilst some progress has been made (thanks for your efforts), unfortunately I'm still not totally convinced on either of these accounts. The relationship to conventional learnability is still only described heuristically and with brief experiments. Whilst the interpretation of the Transformer bound is better and I can see that this method might (roughly) capture a necessary condition for learnability, I think there is still limited evidence of any really practical conclusions or outcomes. For these reasons, I'll stick with my current score, but will be willing to reconsider if the reviewers can convince me I'm wrong on these points before the end of the discussion period.

---

> ### Author Response · Authors · 2024-11-24
>
> We thank the reviewer for taking the time to discuss our revised version and response. Below we address the reviewer’s remaining concerns.
>
> *theoretical result*. We can add the following comments (**1**) when $q(x)$ makes some sense for the task at hand the cross-dataset learnability has significance beyond its ability to serve as a proxy for the on-dataset learnability. (**2**) Cross-dataset learnability can certainly be related theoretically to performance on the original distribution $p(x)$ under assumptions on $p(x)$; one such assumption and bound pair are:
> Assuming $p(x)/q(x) \leq C \quad \forall x \in {\rm supp} ~  p$ the MSE error on $p(x)$ is bounded from below
> $$
> \frac{1}{C}\sum_{i}(L_{i}^{q}-1)^{2}\left\langle \phi_{i},y\right\rangle_q ^{2}=\frac{1}{C}\int dxq(x)[f(x)-y(x)]^{2}\leq\int dxp(x)[f(x)-y(x)]^{2}.
> $$
> This is a concrete bound on the generalization error on the data distribution of interest $p(x)$.
>
> *Practical implications.* We addressed both natural language and vision domains in non-toy-setting scenarios. Moreover, our own experimental results encompass three real dataset. Considering many theoretical works use synthetic or hand-crafted data and stylized neural networks the authors believe practical significance is not a weak point of the current theoretical work.
>
> We agree there are many interesting directions for future research and we believe this is a good indication that this work and the concept of cross-dataset learnability will be of much interest to the community.

---

> > ### Author Response · Authors · 2024-11-28
> >
> > We would like to follow up on this and see if our experiments, revisions, and comments answered the questions and concerns posed by this reviewer. Please let us know if any further clarifications are needed or if any concerns remain.

---

> > > ### Comment · Reviewer_LiQC · 2024-11-28
> > >
> > > Thanks for the response.
> > >
> > > Yes, this is the type of theoretical bound I had in mind for relating cross-dataset learnability, conventional learnability, and generalisation error on a data distribution. I think incorporating this into the manuscript and providing more thorough empirical investigation, perhaps also under different/more sophisticated assumptions on the relationship between $q(x)$ and $p(x)$, would be a big future improvement. I notice 3E5a had similar concerns.
> > >
> > > Likewise, I appreciate the author's efforts to improve the  experimental results, especially upgrading Figs 1 and 2. I'm not sure that Sec. 4.3 can honestly be described as the 'natural language domain in a non-toy setting', and likewise for learning randomly chosen hyperspherical harmonics on PCA'd image datasets, but I do agree that this is moving in the right direction.
> > >
> > > On balance, I'll raise my score from 5 to 6 to reflect the improvements. But I'm still not totally convinced this is publication-ready so will decrease my confidence from 3 to 2.

---

> > > > ### Author Response · Authors · 2024-12-03
> > > >
> > > > We thank the reviewer for the response and for helping us improve both the presentation and the focus of the paper's experiments.
> > > >
> > > > We will certainly consider providing more experiential results. Specifically, we will polish the figures and experiments in the appendix that use the true binary classification target rather than random features. We would also consider adding figures where we use the kernel on top of learned features (e.g., from a pre-trained CNN).
> > > >
> > > > We agree that copying head, a prerequisite for induction heads, is an extremely simple function, nevertheless, we do not view it as a toy setting as it allows us to address a puzzle that naturally appeared in an empirical paper. Since Olsson et al., (2022) a rich literature has evolved around induction heads and their appearance (for example [1-5]).
> > > >
> > > > Finally, we would consider proving the relationship between the two learnabilities under more sophisticated assumptions. We hope further experiments will also help bridge the gap between these two concepts.
> > > >
> > > > [1] Edelman B L, Edelman E, Goel S, Malach E and Tsilivis N 2024 The Evolution of Statistical Induction Heads: In-Context Learning Markov Chains Online: http://arxiv.org/abs/2402.11004
> > > >
> > > > [2] Nichani E, Damian A and Lee J D 2024 How Transformers Learn Causal Structure with Gradient Descent Proceedings of the 41st International Conference on Machine Learning International Conference on Machine Learning (PMLR) pp 38018–70 Online: https://proceedings.mlr.press/v235/nichani24a.html
> > > >
> > > > [3] Bietti A, Cabannes V, Bouchacourt D, Jegou H and Bottou L 2023 Birth of a Transformer: A Memory Viewpoint Online: http://arxiv.org/abs/2306.00802
> > > >
> > > > [4] Reddy G 2023 The mechanistic basis of data dependence and abrupt learning in an in-context classification task The Twelfth International Conference on Learning Representations Online: https://openreview.net/forum?id=aN4Jf6Cx69
> > > >
> > > > [5] Singh A K, Moskovitz T, Hill F, Chan S C Y and Saxe A M 2024 What needs to go right for an induction head? A mechanistic study of in-context learning circuits and their formation Online: http://arxiv.org/abs/2404.07129

---

### Official Review · Reviewer_gVph · 2024-10-27

**Soundness:** 3
**Presentation:** 2
**Contribution:** 2
**Rating:** 6
**Confidence:** 2

**Summary:**

Using the eigenpairs generated under an ideal probability measure, the authors provide a lower bound for the sample complexity for the learnability of the dataset generated from another probability distribution under the KRR and GP framework.

**Strengths:**

1. Clear experiments to explain the main result.
2. Provide applications/examples for the main result in Section 4.
3. The idea of using eigenpairs from an ideal measure and applying it to real data is rather new to me (although I'm not familiar with the literature in this area, and it may be a common technique).

**Weaknesses:**

In general, I’m not familiar with this topic as my area is more in classical kernel methods; therefore, please use my comments (not weaknesses) below sparsely.

1. I’m not familiar with the kernel associated with the neural network (or many of its variances with specified structures, like a transformer or CNN). However, I want to get a general understanding of the issue raised by the non-symmetric data is specific to this NN-related symmetric kernel. Since in classical kernel methods, there seems no need to consider data symmetric, but in the NN context, uniform distribution for covariate is common. Can authors explain more about the challenges (technically) and consequences presented by non-symmetric data?
2. Why is cross-data learnability considered to take the form of Eq. (6)?  How does it link to the original definition of Eq. (4)? I can't see a clear connection to the Eq. (4),  there should be a clearer explanation for this consideration.
3. It looks like the learnability for a feature defined in Eq. (4) takes Tikhonov regularization, which indeed belongs to a broader class called spectral algorithms or spectral regularized algorithms. I wonder if it is possible to extend the result to this class. If not, what is the main obstacle?

**Questions:**

1. Although it is a theoretical bound, I’m curious about its role in practice; how do you estimate the expectations in Eq. (7), especially for those expectations over $D_{P}$. If one uses the empirical counterparts, it is weird as the r.h.s. of Eq. (7) also concerns sample size.

2. Notations
    a. Isn’t the index for $y(x_{\rho})$ and $k(x,x_{\nu})$ should stay the same?
    b. What is the name of quantity in Eq. (6), cross-data learnability?

---

> ### Author Response · Authors · 2024-11-20
>
> We thank the reviewer for taking the time to review our work and providing thoughtful comments. We further thank the reviewer for recognizing the novelty of our work, indeed we believe this is the first time the idea of using an idealized measure to understand real data is introduced in this form. Regarding the specific points and questions
>
> - *...Can authors explain more about the challenges (technically) and consequences presented by non-symmetric data?*
>
> **Response:** Indeed many theoretical works in the NN community assume uniform data and this assumption facilitates the theoretical analysis, however, these assumptions rarely hold for real-world data. Our work capitalizes on the fact the symmetry of the kernel is a far more valid assumption than the symmetry of the data.
>
> Specifically, for the case of kernel regression, If one wishes to give theoretical predictions one has to solve an eigenvalue problem for the kernel on the data to estimate the learnability for example. In general, for real-world data, this task cannot be carried out analytically. One way of accomplishing this task is by relying on symmetries, but these have to be respected by both the kernel and the dataset (see Def. 3.2,3.3). This fact is especially “disappointing” considering that kernels that arise as the limit of neural networks often have large symmetries (that is, the group of symmetries they respect is large). If the symmetry is not a dataset symmetry, naively, it cannot be used to simplify the eigenvalue problem; this work shows it can still be used, but in that case, the results are weaker.
>
> We hope we have answered the point raised by the reviewer but we are not certain. If this is not the case we would be happy to clarify any further issue.
>
> - *Why is cross-data learnability considered to take the form of Eq. (6)? How does it link to the original definition of Eq. (4)?*
>
> **Response:** We thank the reviewer for drawing our attention to this issue. We agree the paper can greatly benefit from such an explanation. First, we add to the definition in equation (4) another form of the quantity which makes the connection much more apparent. We also added an explanation addressing this issue to the paper below Eq. 6. We give it here:
> >“Comparing equations 4,6 the inner product (equivalently expectation) in the cross-data learnability is taken over the auxiliary measure instead of on the dataset.
> >This change can be intuitively understood as learning from $D_P$ but judging how good the reconstruction is based on functional similarity on $q(x)$. This choice can have positive effects, like weighting all possible inputs evenly, perhaps capturing a notion of out-of-distribution generalization; the issue of judging similarity differently based on different measures is exemplified and discussed further in Section 4.2.
> >Cross-dataset learnability agrees with the common one when learning from the true underlying probability density of the dataset $p(x)$ (rather than the empirical $D_P$) and choosing $q(x) = p(x)$. However, The underlying density $p(x)$ is often inaccessible, in these cases, one can still use our generalized learnability to estimate learnability from the empirical dataset $D_P$.
> >We note that while the on-dataset learnability in Eq.4 is bounded $\mathcal{L}_t \in [0,1)$, the cross-dataset learnability is unbounded from above. As a consequence, maximizing cross-dataset learnability does not imply good learning; instead, one must require it to be close to unity.”

---

> ### Author Response · Authors · 2024-11-20
>
> - *It looks like the learnability for a feature defined in Eq. (4) takes Tikhonov regularization...*
>
> **Response:** This is an interesting question. The regularizer is indeed essential in our derivation as it allows us to bound the spectral norm of $[K+I\sigma^2]^{-1}$, where the kernel is evaluated on the dataset $D_P$. While none of the authors is an expert on spectral regularization, it seems the most straightforward way to extend the results of this paper is bounding the spectral norm of the spectral filter used in the spectral regularized algorithm. For Tikhonov the answer is extremely simple, it is the inverse of the ridge, for Landweber iterations it seems the spectral norm of the filter would scale linearly with the number of iterations (see for example [1]).It seems truncated SVD regularization would require knowledge of the spectrum of the kernel evaluated on the dataset $D_P$, which is a requirement we would like to avoid.
>
> We add two remarks. (1) Tikhonov regularization is well-motivated in the context of kernels from neural networks as it relates to the noise in the gradients (Naveh et al., 2021). (2) This regularization arises naturally when learning from an empirical dataset rather than a distribution as shown in Canatar et al., (2021), Simon et al., (2023).
>
> - *Although it is a theoretical bound, I’m curious about its role in practice; how do you estimate the expectations in Eq. (7), especially for those expectations over. If one uses the empirical counterparts, it is weird as the r.h.s. of Eq. (7) also concerns sample size.*
>
> **Response:** We thank the reviewer for the question and now mention the answer in the text explicitly. Indeed one would calculate it empirically, this should be relatively cheap computationally (much cheaper than inverting a kernel). The bound will indeed fluctuate a little as one changes P and more samples are accounted for. As an alternative, one can calculate it on the full dataset and scale $P$ alone. Moreover, This can often be estimated analytically given that some target functions have a simple out structure, e.g.  classification targets usually have norm 1.
>
> - *Notations a.*
>
> **Response:** In general, this form makes sense as is, as the spectral filter (inverse of the kernel with ridge) would normally connect the two quantities by matrix multiplication, these indices just spell out this matrix multiplication (see for example equation 1). We hope we have answered the point the reviewer was referring to, if not we would be happy to clear this matter further.
>
> - *Notations b.*
>
> **Response:** Indeed this quantity is dubbed cross-dataset learnability.
>
> [1] L. Rosasco. Lecture 7 of the Lecture Notes for 9.520: Statistical Learning Theory and Applications. Massachusetts Institute of Technology, Fall 2013. Available at https://www.mit.edu/~9.520/fall13/slides/class07/class07_spectral.pdf

---

> ### Comment · Reviewer_gVph · 2024-11-26
>
> I thank the authors for the detailed replies to my questions, which helped me understand the paper better. After reading other reviewers' comments, I think they have more expertise on this topic than I do, so I decided to keep my score and confidence.
>
> One additional comment about extending the result to spectral algorithms is that: if the authors need to bound the spectral norm of $[K+\sigma^{2}I]^{-1}$ (or its counterpart in spectral algorithms $\phi_{\lambda}(K)$ where $\phi_{\lambda}$ is the filter function), this can be directly done by directly using the properties of the filter function, which leads to the spectral norm bounded by $\lambda^{-1}$. I hope this can provide some useful insights if the authors would like to expand the content of this paper.

---

### Official Review · Reviewer_3E5a · 2024-11-04

**Soundness:** 2
**Presentation:** 2
**Contribution:** 2
**Rating:** 3
**Confidence:** 4

**Summary:**

The paper considers Kernel Ridge Regression in the setting where the set of input points in the training dataset is different and possibly unrelated to the empirical distribution used to construct the eigenfunctions of the model. It is argued that this setting allows to apply spectral results available in a symmetric setting to non-symmetric datasets. The paper first introduces a cross-dataset concept of learnability of an eigenfunction and establishes a lower bound on the sample size required to achieve this learnability. Then, this bound is applied in a series of different settings: a toy gaussian example, the parity function on a hypercube, and learning copying heads with transformer.

**Strengths:**

This is a theoretical paper that attempts to develop an original approach to the study of learnability based on considering it with respect to
a symmetric distribution and kernel but when performed on non-symmetric data. The paper proposes an associated concept of cross-dataset learnability. The main theorem is a lower learnability bound (Theorem 3.1); its proof, provided in the appendix, is concise and correct. The paper discussses several possible applications of this bound to various learning scenarios across different types of targets (gaussian, parity function, copying map).

**Weaknesses:**

1. My main issue with the paper is that I don't see it resulting in clear and important new findings. To begin with, while the paper is theoretical, the only result stated as a theorem is Theorem 3.1, which is a fairly simple bound and seems to be of auxiliary nature. The rest of the paper looks like a mathematically not well organized collection of various calculations and observations.

2. There are some issues with the proposed definition (6) of cross-dataset learnability. The original learnability (4) is a value between 0 and 1 and is an increasing function of sample size $P$ that converges to 1 in the limit $P\to\infty$. In this case there is a clear connection between learnability and sample size, the target is always underlearned, and making learnability $\epsilon$-close to 1 amounts to making $P$ larger than a particular threshold.
In contrast, the cross-dataset learnability $\mathcal L^{D,q}_t$ defined by Eq. (6) can easily be greater than 1. For example, if the target $y$ is an eigenfunction $\phi_s,$ then for $t\ne s$ we have $\langle \phi_t,y\rangle_q=0,$ but in general $\langle \phi_t, \hat f_D\rangle_q\ne 0$ (say if $D=\\{x_1\\}$, then Eq. (29) implies $\langle \phi_t, \hat f_D\rangle_q = \lambda_t (k(x_1,x_1)+\sigma^2)^{-1}\phi_t(x_1)\phi_s(x_1),$ which is nonzero in general). Moreover, there is no obvious connection between the learnability  $\mathcal L^{D,q}_t$ and the dataset size. The learnability can stay above and away from 1 even in the limit $P\to\infty$. Due to this possibility of overlearning and contrary to what is suggested in and around Theorem 3.1, the condition $\mathcal L^{D,q}_t>1-\epsilon$ does not imply that we accurately learn the respective eigencomponent. These issues are not discussed in the paper.

3. I cannot say that the paper convincingly demonstrates the importance and usefulness of the proposed approach to analysis of learning based on the cross-dataset learnability  $\mathcal L^{D,q}_t$ and Theorem 3.1. As seen e.g. in Figure 1, actual learnability can deviate significantly from the derived bound, and the paper provides no results on its tightness. The vignettes considered in Section 4 are not that convincing, either. The first, Gaussian vignette is a very toy example. In the second, an existing exponential complexity bound for learning the parity function is shown to hold for non-symmetric distributions, but it's not clear why we should be interested in non-symmetric distributions in this problem. In the third vignette on copying heads, I also don't understand the significance of the resulting bound (26); there is no discussion on that in the paper.
While the presented research is described as motivated by the practical problem of assessing learnability on real-world data, I don't see any useful applications to such data presented in the paper. For example, Figure 1 shows some results related to CIFAR-10, but they  seem to be very far from answers to real practical questions such as "how big must be the data set to achieve accuracy x?"

4. The paper has a number of typos and confusing statements:

   * line 705: "*while the eigendecomposition of the operator $K$ ... depends on the measure $q(x)$, the Mercer decomposition of the kernel function $k(x, y)$ does not*" - that's not correct, Mercer's decomposition does depend on the measure (otherwise how would we define orthogonality?)

   * Line 174: "*The learnability computed empirically divided by our learnability bound*"- The bound established in Theorem 3.1 is for the number of samples, not the learnability.  Also, it is not clear to me what is $B$ in Eq. (10), and how this equation was derived.

   * $\phi_1$ should be $\phi_2$ in the left panel of Figure 1

   * The numerator in Eq. (7) seem to be missing the modulus

   * Line 712: the reference to Eq. (5) should probably refer to Eq. (1)

   * Line 396: "Sterling’s formula" - Stirling’s formula

   * Line 451: "casual masking" - probably, causal

In summation, I don't think that the paper is ready for publication in the present state.

**Questions:**

-

---

> ### Author Response · Authors · 2024-11-20
>
> We sincerely thank the reviewer for the time spent reviewing and commenting on our work. These comments have been valuable to us and enabled us to sharpen the messages significantly. We further thank the reviewer for recognizing the novelty in the approach taken in this work. We hope the revision made to the paper communicates the results better and can convince the reviewer that cross-dataset learnability is not only an interesting new quantity but also a useful one. Below we address the reviewer's specific concerns.
>
> - *My main issue with the paper is that I don't see it resulting in clear and important new findings.*
>
> **Response**: We thank the reviewer for pinpointing this issue. One of the main theoretical contributions of this work is the definition of cross-dataset learnability and the demonstration of its tractability, this is now complemented by better experiments that show its usefulness. The fact that this quantity can be bounded by a straightforward argument is not a disadvantage in our view. Additionally, while the derivation is rather direct, to the best of our knowledge, the result is novel. Regarding the significance of the results, we believe the new experiments and the rest of our answers address this concern, if this is not the case, we would be happy to discuss it further.
>
> - *There are some issues with the proposed definition (6) of cross-dataset learnability...*
>
> **Response**: We agree with the reviewer. We would like to apologize for a typo in the main text, one should require $L=1-\epsilon$, as in the last equation in Appendix A, rather than $L \geq 1-\epsilon$, Eq. (7) (now Eq. 8) then follows. The requirement $L=1-\epsilon$ does indeed rule out the case of overlearning. We would also like to note we did not claim the cross-dataset learnability is bounded by 1. We now state this explicitly in the text to avoid this misunderstanding.
>
> Regarding the specific example presented by the reviewer, we note our bound in Eq.(7) (now 8) gives $P^* \geq 0$ which holds trivially. We stress our bound is a necessary, but not a sufficient condition for learning, i.e. an upper bound on the (cross-dataset) learnability. There are indeed cases where our bound does not prove useful, but it is able to give relevant sample complexity bounds in real-world scenarios, we show evidence for that in new figures that we have added to the paper (see next point).
>
> - *I cannot say that the paper convincingly demonstrates the importance and usefulness*
>
>
> **Response**: We thank the reviewer for bringing up this point. We believe it has greatly helped us in presenting the results of the paper. According to the comments by the reviewers, we see the figures were ineffective in communicating the message of the paper. Therefore we decided to perform new experiments and add new figures while delegating the existing ones to an appendix.
>
> The new experiments compare the regular learnability, the cross-dataset learnability, and our sample complexity bounds. We see these as evidence for our sample complexity bound as an indicator of when learning occurs.
>
> To comment on the specific point, the old Figure 1, left panel, is actually rather tight during the learning phase (to the left of the horizontal line) allowing us to give good predictions for the sample complexity. This can be seen by the fact the ratio of the bound to the learnability is always within an order of magnitude during the learning phase.  The old Figure 2, left panel, also shows the bound is within an order of magnitude from the exact learnability throughout the learning phase. The bound naturally becomes very loose as it grows linearly with $P$ even after the learnability saturates, but again our aim is to give a necessary condition for learnability, below a certain complexity some functions cannot be learned.

---

> ### Author Response · Authors · 2024-11-20
>
> - *The vignettes considered in Section 4 are not that convincing...*
>
> **Response**:  The vignettes serve a double purpose in this manuscript, (1) exemplifying how the bound can be used and shedding light on its different aspects, while also (2) providing some new results. The Gaussian vignette serves mainly the first purpose. Regarding the parity vignette, parity is used to model a high-degree, complicated, feature that is shown to be unlearnable in image classification for instance with FCN kernels using our bound. Moreover, although the ML folklore says high-degree polynomials such as parity (in the noiseless case) are hard to learn from data by FCN-kernels, we are not aware of a result that shows it in any case but under strong symmetry assumptions of the distribution.
>
> Concerning the copying heads vignette, induction heads (Olsson et al., 2022) were shown empirically to be responsible for some of the in-context learning in LLMs. In turn, learning copying heads is a necessary step in learning induction heads, and thus a necessary step for a fundamental in-context learning capability. This case is brought forth as an example where we have a simple closed-form expression for a target function that has relevance in real-world performance. The result in the paper is a concrete lower bound on the sample complexity of in-context learning induced by induction heads for transformers in the lazy regime. Specifically, the result shows a form of “curse of dimensionality” where the sample complexity scales with the product of the context and the vocabulary. We note that this is also seen in a range of real-world large transformers outside the kernel regime. Olsson et al., (2022) consistently found it takes more than a billion tokens before induction heads are learned, this is a non-trivial amount. A billion tokens are approximately 10 times more than the large version of WikiText or 1% of the gigantic C4 dataset.
>
> - *For example, Figure 1 shows some results related to CIFAR-10, but they seem to be very far from answers to real practical questions such as "how big must be the data set to achieve accuracy x?"*
>
> **Response**:  The reviewer is asking for an extremely challenging type of prediction that involves real-world datasets with all their complexity together with networks as complicated as transformers. Still, we believe we rise to this challenge. Given one knows the target function we give the user useful predictions. Our cross-dataset learnability is shown (in the new figures) to be of the same order of magnitude as the on-dataset learnability which is directly related to the mean square error. As aforementioned we can do that for real-world and complicated networks at the kernel regime.
>
> - *line 705: "while the eigendecomposition of the operator... depends on the measure, the Mercer decomposition of the kernel function does not" - that's not correct, Mercer's decomposition does depend on the measure (otherwise how would we define orthogonality?)*
>
> **Response**:  We thank the reviewer for helping us make this statement more precise. This statement was meant to communicate the fact that the Mercer representation found on one dataset can be used on another (given the condition on their support). We agree with the reviewer that the statement, taken at face value, is incorrect and have rephrased it to “the representation of the kernel found by Mercer decomposition holds for all datasets within the support on q”
>
> - *Line 174: "The learnability computed empirically divided by our learnability bound"- The bound established in Theorem 3.1 is for the number of samples, not the learnability. Also, it is not clear to me what is in Eq. (10), and how this equation was derived.*
>
> **Response**:  The reviewer's comments made clear the experiments were not communicated well enough, we are sincerely thankful for this input. This figure is now delegated to an appendix, where we give a better explanation for what we depict in it. To give a short answer, we did not use the sample complexity bound, but the bound on the cross-dataset learnability derived in the appendix, we can see how that can be confusing, and we now present things differently. We hope the reviewer will be able to comment on the new figures too.
>
> - *Typos*
>
> **Response**:  We thank the reviewer again for catching all of these! We apologize for any inconvenience they may have caused during the reading.

---

> ### Author Response · Authors · 2024-11-28
>
> We would like to follow up on this and see if our new experiments, revisions, and comments answered the questions and concerns posed by this reviewer. Please let us know if any further clarifications are needed or if any concerns still remain.
>
> We would also like to highlight that under assumptions on the distribution of interest $p(x)$ the cross-dataset learnability can be related to the performance on $p(x)$. For example, assuming $p(x)/q(x) \leq C \quad \forall x \in {\rm supp} ~  p$ the MSE error on $p(x)$ is bounded from below
> $$
> \frac{1}{C}\sum_{i}(L_{i}^{q}-1)^{2}\left\langle \phi_{i},y\right\rangle_q ^{2}=\frac{1}{C}\int dxq(x)[f(x)-y(x)]^{2}\leq\int dxp(x)[f(x)-y(x)]^{2}.
> $$

---

> > ### Comment · Reviewer_3E5a · 2024-11-29
> >
> > I thank the authors for their response. But I still find the new version hard to read and even confusing.
> >
> > I even find the statement of the only theorem in the paper confusing. The bound (8) there is formulated as if there were two different $P$'s: one provided with the set $D_P$ and another for which the bound $P\ge P^*$ holds. But there is just one $P$, introduced in the beginning of the theorem. Formula (8) is apparently derived from formula (7) which has the structure $\mathcal L\le Pf(P)$ with some function $f(P)$. If we assume $\mathcal L\ge 1-\epsilon,$ this implies the condition $P\ge (1-\epsilon)/f(P).$ This is a condition on the original $P$ we started from; I don't understand the point of introducing the new object $P^* = (1-\epsilon)/f(P)$. Yes, the bound $P\ge P^*$ would need to hold, but only for the particular $P$ for which $P^*$ is defined.
> >
> > This confusing double meaning of $P$ seems to also be present in some other places. In particular, the description of Figure 1 in line 251 says that "*the dataset $D_p$ is $10^4$ samples*". I understand this as saying that $P=10^4$. But what then stands in the "Trainset Size ($P$)" axis in Figure 1?
> >
> > There is a related issue even earlier, in formula (5). The eigenvalues $\lambda_t$ are defined in formula (2) as the eigenvalues of the kernel on the dataset $D_P$. In particular, the eigenvalues $\lambda_t$ depend on $D_P$. Then, in formula (5) the expression defining $P_t^*$ implicitly depends on $P_t^*$ itself.
> >
> > In fact, even the notation $P_t^*$ here is somewhat confusing, because it uses the single index $t$ as if uniquely identifying the eigenvalue, whereas in general the eigenvalues $\lambda_t$ for one dataset $D$ are unrelated to the eigenvalues $\lambda'_s$ for another dataset $D'$.
> >
> > Also, an obvious issue with the proposed learnability expression (6) is that it is defined as an absolute value of a fraction that can be, for example, equal to -1. In this case the learnability $\mathcal L_t^{D,q}=1$, which the paper seems to view as the most desirable outcome. But it is clear that the learnability in the intuitive sense in this case is, conversely, very poor. I understand that the authors focus on the necessary rather than sufficient conditions of learning, but this possibility shows that the interpretation of the expression $\mathcal L_t^{D,q}$ introduced in (6) as learnability can just contradict common sense.

---

> > > ### Author Response · Authors · 2024-12-01
> > >
> > > Addressing first the matter of $P$ vs $P^*$, there is indeed only one $P$. The notation $P^*$ was meant to ease the reading and be reserved for sample complexity, apparently, this was not the case. We will consider removing this notation in future versions.
> > >
> > > To be concrete, in our Eq. (8), applied for instance to CIFAR-10's full training set, one would choose a feature of interest ($\phi_t$) and compute the right-hand side w.r.t. to $D_P={\rm CIFAR}\_{\rm TRAIN}$. If $P^* < 50,000$ (CIFAR train set size) the feature cannot be learned to the given accuracy. Taking subsets of CIFAR-10 train set, the value for $P^*$ would indeed depend on the particular realization(i.e. the random choice of samples to be included in the training) however in a random way rather than a $P$ dependent way. We do not see any logical problem with this, it is simply a $D_P$ dependent quantity as one may expect. Also, note in this respect that the quantities in the r.h.s. of Eq.(8) that depend on $D_P$ are computationally cheap to evaluate - $O(P)$. Significantly cheaper than KRR which scales as $O(P^3)$. Finally, it is worth mentioning the r.h.s. depends on the training set realization but it does not scale with $P$, in fact for large $P$ values we would expect it to fluctuate very little between realizations based on the law of large numbers. Our experiment in Fig.1 left panel supports that, as one can see different realizations come closer together as $P$ increases, and become indistinguishable around $P=2000$.
> > >
> > > We agree with the reviewer that the absolute value should not have been present in the definition of learnability in Eq. (6) (it has no effect in e
> > > Eq. (4)); we sincerely thank the reviewer again for raising this point. The main results remain unchanged by this change in definition if one adds the assumption that $\langle \phi_t, y \rangle \geq 0$ without loss of generality (the sign of $\phi_t$ can always be chosen to satisfy this condition).

---

### Official Review · Reviewer_Tb3N · 2024-11-04

**Soundness:** 3
**Presentation:** 2
**Contribution:** 2
**Rating:** 5
**Confidence:** 5

**Summary:**

The authors study KRR via spectral analysis of their eigenvalues  on the input
data.
They argue while kernels are more symmetric than data, one may still use eigenvalues and eigenfunctions associated
with idealized data to provide bounds on realistic data via distributional shifts. As an example, they give a theoretical lower bound on the sample complexity of
copying heads for kernels associated with generic transformers acting on natural language, along with supporting experiments.

**Strengths:**

- study of the role of symmetries in KRR
 - having examples

**Weaknesses:**

- the contribution of the paper is not clear; I didn't understand it by reading the paper
 - the intro section does not provide anything about the paper because it is too general
 - missing references

**Questions:**

This is an interesting paper. I think this paper needs revision to make its contributions more clear, along with a comparison with previous works. Here are my comments and questions:

 - The introduction section is not clear--there are a lot of generic arguments, including all those in the 'our contributions' part (e.g., 'We generalize ... is tested on another.'). One cannot understand what exactly the contribution of the paper is based on the introduction section. This, in my opinion, should be clarified and revised.

- typo in equation 1: comma is missing in the summation

- Equation 2 is not clear--are \psi functions vector? Then what does $[\psi_i,\psi_j]$ mean? Is that $L^2$ inner product? The paper would benefit from a more clear notation.

- In Equation 2, can you explain why the expectation is over the data set, not data distribution?

- The setup considered in Section 2 (Equation 6) is called the study of distributional shift, and some recent works are missing in references such as [1]. Moreover, recent works are missing for learning kernels with symmetries [2,3]. These are just instances, and please find similar papers for reference.

 - Theorem 3.1 does not deal with symmetry, and the role of the proposed workflow is unclear to me.

I'm open to changing my score depending on the authors' response.




[1] Ma, Cong, Reese Pathak, and Martin J. Wainwright. "Optimally tackling covariate shift in RKHS-based nonparametric regression." The Annals of Statistics 51.2 (2023): 738-761.

[2] Bietti, Alberto, Luca Venturi, and Joan Bruna. "On the sample complexity of learning under geometric stability." Advances in neural information processing systems 34 (2021): 18673-18684.

[3] Tahmasebi, Behrooz, and Stefanie Jegelka. "The exact sample complexity gain from invariances for kernel regression." Advances in Neural Information Processing Systems 36 (2023).

---

> ### Author Response · Authors · 2024-11-20
>
> We thank the reviewer for investing the time in reviewing our paper and providing constructive feedback. We further thank the reviewer for keeping an open mind regarding our future response. Below we respond to the specific points.
>
> - *the contribution of the paper is not clear; I didn't understand it by reading the paper*
> - *the intro section does not provide anything about the paper because it is too general*
> - *The introduction section is not clear--there are a lot of generic arguments, including all those in the 'our contributions' part (e.g., 'We generalize ... is tested on another.'). One cannot understand what exactly the contribution of the paper is based on the introduction section. This, in my opinion, should be clarified and revised.*
>
> **Response:** We thank the reviewer for their honest opinion. We made several changes to the text, including sharpening the “main contributions” and working out through the introduction, hopefully removing ambiguities and presenting points more clearly. We also performed new experiments which we believe make the overall presentation more clear and cohesive.
>
> - *typo in equation 1: comma is missing in the summation*
> - *Equation 2 is not clear--are $\psi$ functions vector? Then what does mean$[\psi_i,\psi_j]$? Is that
>  inner product? The paper would benefit from a more clear notation.*
>
> **Response:** We thank the reviewer for taking the time to help us clear such typos. We added the comma in equation 1 and removed the comma in equation 2 this is just an expectation over a product of two psi’s. One can indeed define an inner product with this expectation over products but we chose not to, to avoid introducing unnecessary notation. This is now consistent with our definition of cross-dataset learnability which uses the same expectation over a product notation instead of an inner product. The introduction of the inner product is delegated to the appendix.
>
> Regarding the status of psi, it is a function as suggested by the notation, and as given by Mercer’s theorem. In equation 2 this function is restricted to the dataset $D_P$ (notice this restriction on the rightmost part of equation 2). This notation unifies the treatment for both discrete and continuous distribution, which may also be relevant to the reviewer’s next point.
>
> - *In Equation 2, can you explain why the expectation is over the data set, not data distribution?*
>
> **Response:** We thank the reviewer for the question. If the “data distribution” in the question is to be interpreted as an empirical data distribution (say sum of delta distributions) then the two are the same. If, on the other hand, the reviewer means the “theoretical” underlying distribution, e.g. uniform measure on the hypersphere compared to P points sampled uniformly on the hypersphere, then in real-world scenarios this distribution is often unknowable -  we do not know the true underlying distribution of images of handwritten digits for example. Defining the learnability on the dataset allows us to treat such real-world examples.
>
> - *The setup considered in Section 2 (Equation 6) is called the study of distributional shift, and some recent works are missing in references such as [1]. Moreover, recent works are missing for learning kernels with symmetries [2,3]. These are just instances, and please find similar papers for reference.*
>
> **Response:** We thank the reviewer for bringing up this interesting point. The setup in equation 6 indeed resembles that of a distributional shift but we would like to draw several distinctions between them. In a distributional shift, one learns from one dataset but would ideally want to learn from another, the question is then: given we learned from the “wrong” dataset what can be said about performance on the “correct” one. Instead, this work assumes detailed knowledge of the would-be “shifted” distribution $q(x)$, not the training dataset $D_P$. Furthermore, In this work, the distribution $q(x)$ has only an auxiliary nature. One has the freedom to handcraft a simple and analytically tractable $q(x)$ so quantities like learnability can be analytically calculated on it. Nevertheless, this is an interesting connection and we now mention it in a new appendix on further related works.
>
> Regarding refs. [2,3]; we would like to thank the reviewer again for broadening the scope of the discussion. While our references so far discussed symmetries that appear naturally in the architecture of neural networks the two papers mentioned by the reviewer refer to the case where one imposes more symmetries. This is indeed an interesting topic as well and we included it in the above-mentioned new appendix on further related works.

---

> ### Author Response · Authors · 2024-11-20
>
> - *Theorem 3.1 does not deal with symmetry, and the role of the proposed workflow is unclear to me.*
>
> **Response:** The reviewer is correct, theorem 3.1 holds regardless of any symmetry considerations. However, one would likely find very little use for it if the distribution $q(x)$ does not yield a simpler kernel eigenvalue problem. Here is the relevant use case: one is interested in the performance on $D_P$, but solving the eigenvalue problem on $D_P$ proves intractable. In this case, we suggest solving it on a symmetric auxiliary distribution $q(x)$, making the solution practical.
>
> Symmetries enter the picture as a tool to simplify kernel eigenvalue problems, but they can be used only when the data distribution obeys them too. Our work, and specifically theorem 3.1 gives a sample complexity bound when learning from a dataset of interest but being able to solve the kernel eigenvalue problem only on a simpler data distribution $q(x)$.
>
> We hope the changes to the introduction will help make this clearer for future readers.

---

> > ### Comment · Reviewer_Tb3N · 2024-11-25
> >
> > I sincerely thank the authors for comprehensively responding to my comments/concerns, and I assure them that I read their rebuttal in detail. For now, I don't have any new particular questions/concerns, and I will decide whether to keep my score or increase it after a bit of necessary discussion with other reviewers/AC. Thanks!

---

### Author Response · Authors · 2024-11-20

We thank all the reviewers for investing the time in reading and commenting on the manuscript. We have uploaded an updated version; specific changes are mentioned in the individual comments for the corresponding issues.

---

### Meta-Review · Area_Chair_wa5r · 2024-12-21

**Metareview:**

This paper considers kernel ridge regression via a spectral analysis of the eigenvalues and eigenfunctions on idealized input. Authors argue that while kernels are often more symmetrical in nature than practical data, one may still use eigenvalues and eigenfunctions associated with idealized data to provide bounds on realistic data via distributional shifts. Authors provide a theoretical lower bound on the sample complexity.

This paper was reviewed by four expert reviewers and received the following Scores/Confidence: 3/4, 6/2, 5/5, 6/2. I think paper is studying an interesting topic but authors are not able to convince the reviewers sufficiently well about the phenomena they are trying to explain. The following concerns were brought up by the reviewers:

- Poor writing and unclear statements: Some technical details are stated in a vague way. Even statement of the theorem is not clear.

- Results in this paper should be better conveyed with a clear conclusion. Authors discussion with some reviewers seem to suggest there is still room for improvement.

- Even some reviewers who suggested accepting the paper are not championing and admitting that the paper is not ready for publicaiton in its currrent state.


No reviewers championed the paper and they are not particularly excited about the paper.
I think majority of the concerns can be addressed but that would require significant revision and another set of reviews. As such, based on the reviewers' suggestion, as well as my own assessment of the paper, I recommend not including this paper to the ICLR 2025 program.

**Additional Comments On Reviewer Discussion:**

Author rebuttal successfully addressed some of the reviewer concerns; yet, after reading the discussion I am convinced that significant work is needed to make this paper ready for publication.

---

### Decision · Program_Chairs · 2025-01-22

Reject